# Dally is not essential for Dpp spreading or internalization but for Dpp stability by antagonizing Tkv-mediated Dpp internalization

**Niklas Simon[1†], Abu Safyan[2,3,4,5,6†], George Pyrowolakis[3,4,5,6*], Shinya Matsuda[1*‡]**

[1]Growth & Development, Biozentrum, Spitalstrasse, University of Basel, Basel, Switzerland; [2]International Max Planck Research School for Immunobiology, Epigenetics, and Metabolism, Freiburd, Germany; [3]Institute for Biology I, Faculty of Biology, University of Freiburg, Freiburg, Germany; [4]CIBSS – Centre for Integrative Biological Signalling Studies, University of Freiburg, Freiburg, Germany; [5]BIOSS – Centre for Biological Signalling Studies, University of Freiburg, Freiburg, Germany; [6]Hilde Mangold Haus, University of Freiburg, Freiburg, Germany

**\*For correspondence:**
g.pyrowolakis@biologie.uni-freiburg.de (GP);
shinyamatsuda0423@gmail.com (SM)

[†]These authors contributed equally to this work

**Present address:** [‡]Department of Biological Sciences, Graduate School of Science, The University of Tokyo, Tokyo, Japan

**Competing interest:** The authors declare that no competing interests exist.

**Abstract** Dpp/BMP acts as a morphogen to provide positional information in the *Drosophila* wing disc. Key cell-surface molecules to control Dpp morphogen gradient formation and signaling are heparan sulfate proteoglycans (HSPGs). In the wing disc, two HSPGs, the glypicans Division abnormally delayed (Dally) and Dally-like (Dlp) have been suggested to act redundantly to control these processes through direct interaction of their heparan sulfate (HS) chains with Dpp. Based on this assumption, a number of models on how glypicans control Dpp gradient formation and signaling have been proposed, including facilitating or hindering Dpp spreading, stabilizing Dpp on the cell surface, or recycling Dpp. However, how distinct HSPGs act remains largely unknown. Here, we generate genome-engineering platforms for the two glypicans and find that only Dally is critical for Dpp gradient formation and signaling through interaction of its core protein with Dpp. We also find that this interaction is not sufficient and that the HS chains of Dally are essential for these functions largely without interacting with Dpp. We provide evidence that the HS chains of Dally are not essential for spreading or recycling of Dpp but for stabilizing Dpp on the cell surface by antagonizing receptor-mediated Dpp internalization. These results provide new insights into how distinct HSPGs control morphogen gradient formation and signaling during development.

## eLife assessment

This **important** study uses genomically-engineered glypican alleles (Dally and Dally-like) to determine the role of these proteins on the Dpp/BMP morphogen gradient in the wing disc of *Drosophila melanogaster*. The new glypican null and tagged add-back alleles, as well as a Dpp mutant that cannot bind heparan sulfate moieties in glypicans, provide **solid** results that support the model in which Dally but not Dally-like stabilizes Dpp on the cell surface by counteracting receptor-mediated Dpp internalization. This paper would be of interest to developmental biologists working on morphogens.

## Introduction

Morphogens are signaling molecules secreted from localized source cells to control cell fates in a concentration dependent manner during development (*Stapornwongkul and Vincent, 2021*; *Tabata, 2001*; *Ashe and Briscoe, 2006*; *Rogers and Schier, 2011*). Amongst them, Decapentaplegic (Dpp), the vertebrate BMP2/4 homolog is the first validated secreted morphogen identified in *Drosophila*. Since its discovery, the role of Dpp signaling in the *Drosophila* wing imaginal disc (the larval precursor of the adult wing) has served as an excellent model to investigate how morphogen gradients form and act (*Affolter and Basler, 2007*; *Matsuda et al., 2016*; *Kicheva and González-Gaitán, 2008*). In the wing disc, Dpp is expressed in an anterior stripe of cells along the anterior-posterior compartment boundary to control nested target gene expression and tissue growth (*Figure 1A*). Dpp binds to the BMP type I receptor Tkv and the BMP type II receptor Punt to induce the phosphorylation of the transcription factor Mad. Phosphorylated Mad (pMad) forms a complex with the co-Smad (Medea) to accumulate into the nucleus and activate or inhibit target gene transcription. Interestingly, the majority of Dpp target genes are repressed by the transcriptional repressor Brinker (Brk), while pMad directly represses *brk* expression (*Campbell and Tomlinson, 1999*; *Jaźwińska et al., 1999*; *Minami et al., 1999*). Thus, the pMad gradient regulates nested target gene expression mainly through generating an inverse gradient of Brk (*Müller et al., 2003*; *Figure 1A*). The nested target gene expression patterns are thought to position the future adult wing veins such as L2 and L5 (*Cook et al., 2004*; *Sturtevant et al., 1997*; *Figure 1A*). In addition to patterning, Dpp signaling controls growth of the wing through repressing Brk as the strong proliferation defects of *dpp* mutant wing discs can be reversed by removing *brk* (*Campbell and Tomlinson, 1999*; *Schwank et al., 2008*).

While endogenous Dpp is indeed distributed in a graded manner (*Matsuda et al., 2021*) how the gradient arises is highly controversial (*Matsuda et al., 2016*; *Kicheva and González-Gaitán, 2008*; *Akiyama and Gibson, 2015*). A class of molecules that control gradient formation of various morphogens are glypicans, GPI-anchored heparan sulfate proteoglycans (HSPGs) consisting of a core protein and covalently attached heparan sulfate (HS) chains (*Lin and Perrimon, 2002*; *Nybakken and Perrimon, 2002*; *Yan and Lin, 2009*). In *Drosophila*, two glypicans, Division abnormally delayed (Dally; *Nakato et al., 1995*) and Dally-like protein (Dlp; *Khare and Baumgartner, 2000*) are thought to redundantly act as co-receptors (*Kuo et al., 2010*), regulate Dpp morphogen gradient formation (*Belenkaya et al., 2004*; *Fujise et al., 2003*), and control scaling of Dpp gradient with tissue size (*Romanova-Michaelides et al., 2022*; *Zhu et al., 2020*).

The role of glypicans in Dpp signaling is dependent on HS chains. A variety of genes required for HS chain biosynthesis such as *sulfateless* (*sfl*) and a family of Ext genes *sister of tout-velu* (*sotv*), *brother of tout-velu* (*botv*), and *tout-velu* (*ttv*) (*Bornemann et al., 2004*; *Han et al., 2004*; *Takei et al., 2004*) have been shown to be involved in Dpp distribution and signaling. Given that Dpp binds to heparin (*Akiyama et al., 2008*), the interaction of the HS chains of glypicans with Dpp is thought to be essential for their function.

Several, not necessarily mutually exclusive, models have been proposed on how glypicans control Dpp morphogen gradient formation. First, glypicans are suggested to be essential for Dpp spreading by handing it over from one cell to the next cell in a 'bucket brigade' manner ('facilitated diffusion' model) (*Kerszberg and Wolpert, 1998*; *Figure 1B*). Consistently, FRAP assays using overexpression of GFP-Dpp revealed a low diffusion coefficient of Dpp (D=0.1 μm²/s) (*Kicheva et al., 2007*). Indeed, pMad signal and Dpp distribution have been shown to be reduced not only in glypican mutant clones but also in wild type cells located adjacent to the clone and distally to the Dpp source (so-called 'shadow' effect), indicating that Dpp spreading requires glypicans (*Belenkaya et al., 2004*).

Second, and in contrast to the first model, glypicans have been proposed to interfere with Dpp spreading (*Müller et al., 2013*; *Figure 1C*). The 'hindered diffusion' model postulates that Dpp spreads freely and transient interaction of Dpp with cell surface molecules such as glypicans hinders effective Dpp spreading (*Müller et al., 2013*).

Third, glypicans have been proposed to stabilize Dpp on the cell surface (*Akiyama et al., 2008*; *Figure 1D*). A previous study showed that Dpp lacking the N-terminal HS binding site is less stable than wild type Dpp (*Akiyama et al., 2008*) indicating that Dally regulates Dpp stability through interactions of HS chains with Dpp. Genetic analyses indicated that Dally competes with Tkv for Dpp binding to antagonize Tkv-mediated Dpp internalization (*Akiyama et al., 2008*).

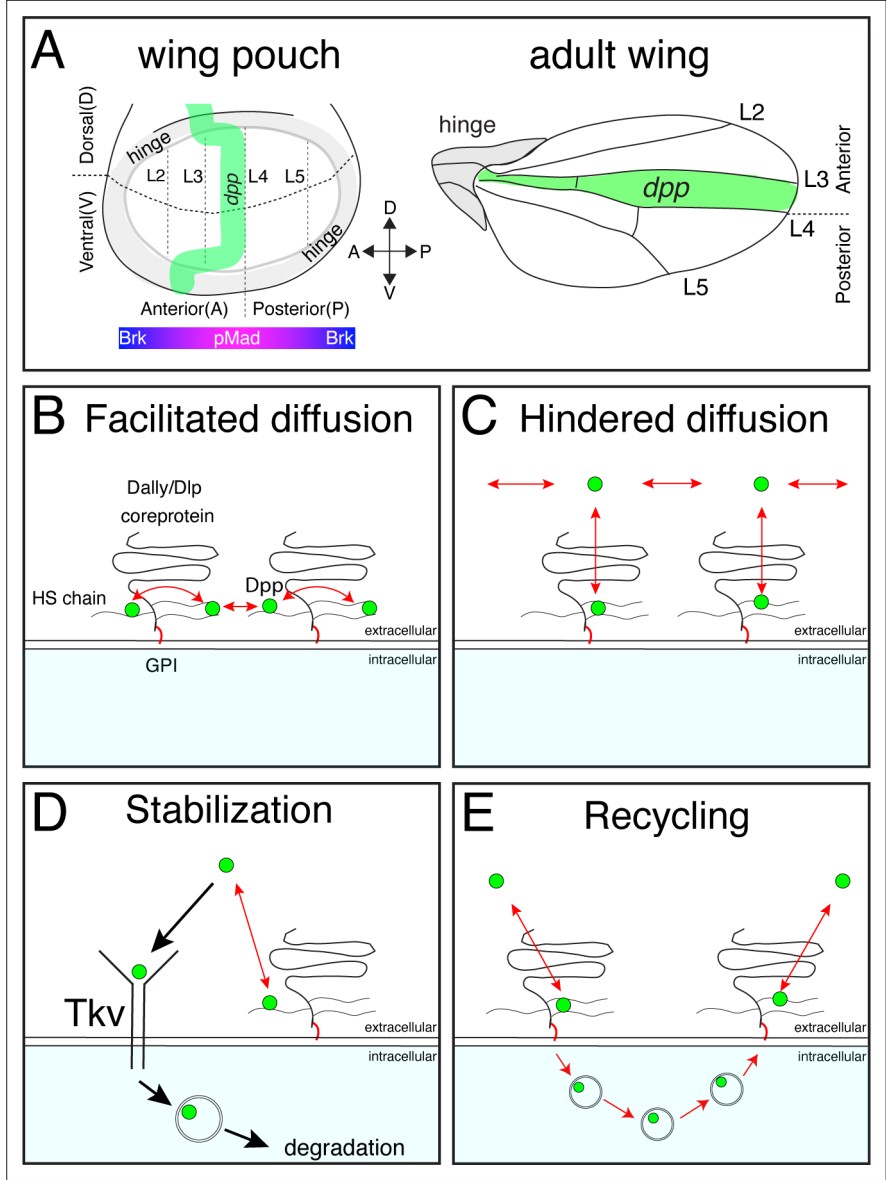

**Figure 1.** Distinct models on the roles of HSPGs on Dpp/BMP gradient formation. (**A**) Schematic view of the wing pouch (future wing tissue) and adult wing tissue. Dpp spreads from the anterior stripe of cells along the A-P compartment boundary to generate pMad gradient and an inverse Brk gradient to specify adult wing veins such as L2 and L5. (**B**) HSPGs are proposed to transport Dpp via repetitive interaction of HS chains with Dpp (facilitated diffusion model). (**C**) HSPGs are proposed to hinder Dpp spreading via reversible interaction of HS chains with Dpp (Hindered diffusion model). (**D**) HSPGs are proposed to stabilize Dpp on the cell surface via reversible interaction of HS chains with Dpp that antagonizes Tkv-mediated Dpp internalization. (**E**) HSPGs are proposed to internalize and recycle Dpp.

Fourth, glypicans are proposed to control recycling and re-secretion of Dpp, which is critical for extracellular Dpp gradient scaling (*Romanova-Michaelides et al., 2022*), a phenomenon in which the Dpp gradient expands to match the growing wing disc size during development (*Figure 1E*). Surprisingly, although receptor-mediated endocytosis is thought to be a mechanism to clear morphogens from the extracellular space (*Figure 1D*), the same study showed that Dpp internalization was not affected in *tkv* mutant clones, indicating that Dally, but not Tkv, mediates Dpp internalization (*Romanova-Michaelides et al., 2022*).

Finally, a recent attempt to create a synthetic morphogen system using free diffusing GFP as a morphogen raises alternative possibilities (*Stapornwongkul et al., 2020*). When *dpp* was replaced

with GFP, and anti-GFP nanobodies fused to Tkv and Punt that can activate pMad signal upon GFP binding were introduced, rescue of severe *dpp* mutant phenotypes was not perfect but improved by further introducing GPI-anchored non-signaling receptors (corresponding to glypicans). Supported by mathematical modeling, this study suggests that glypicans may contribute to morphogen gradient formation by blocking morphogen leakage and assisting morphogen spreading likely via their GPI-anchors (*Stapornwongkul et al., 2020*).

Previously, we generated a platform to introduce epitope tags and/or mutations in the *dpp* locus and protein binder tools to directly manipulate endogenous Dpp spreading (*Matsuda et al., 2021*). By visualizing and manipulating the endogenous Dpp morphogen gradient, we found that Dpp spreading is critical for posterior patterning and growth but largely dispensable for anterior patterning and growth (*Matsuda et al., 2021*). Although the requirements for Dpp spreading were thus less important than previously thought, the endogenous Dpp gradient was easily visualized in the wing discs and Dpp spreading was still critical for posterior patterning and growth (*Matsuda et al., 2021*). Thus, wing imaginal discs serve as an excellent model to study Dpp dispersal-mediated morphogen gradient formation at physiological conditions.

In this study, to better understand how glypicans control Dpp morphogen gradient formation and signaling, we generated genome-engineering platforms to manipulate both *Drosophila* glypicans, Dally and Dlp. Although the two glypicans are thought to act redundantly, we first find that only Dally is critical for Dpp gradient formation through interaction of its core protein with Dpp. Surprisingly, we found that, although the HS chains of Dally are essential for Dally's function, *dpp* mutants that lack the ability to interact with HS chains display only minor phenotypes, indicating that a direct interaction of HS chains of Dally with Dpp is largely dispensable for Dpp gradient formation and signaling. We provide evidence that the HS chains of Dally are not essential for Dpp internalization or spreading but rather stabilize Dpp on the cell surface by antagonizing Tkv-mediated internalization of Dpp. These results provide new insights into how glypicans control dispersal and signaling of distinct morphogens during development.

## Results

### Dally but not Dlp interacts with Dpp

Previous genetic studies suggested that both Dally and Dlp are involved in extracellular Dpp distribution and signaling through the interaction of Dpp with their HS chains (*Belenkaya et al., 2004*). However, the relative contribution of each glypican remained unclear. To address this, we first expressed *dally* or *dlp* in the dorsal compartment of the wing disc using *ap*-Gal4 to compare their ability to interact with endogenous Dpp and activate pMad signal in vivo. To visualize endogenous extracellular Dpp distribution, we utilized a functional *Ollas-dpp* allele (*Bauer et al., 2023*). Since Dpp distribution and pMad signal are similar between the dorsal and ventral compartments in *Ollas-dpp* discs (*Figure 2—figure supplement 1*), the ventral compartment serves as an internal control in this assay. We found that expression of *dally* in the dorsal compartment increased HS chain levels, and expanded the extracellular Dpp distribution and pMad gradient compared to the control ventral compartment (*Figure 2A, B, E and F*). Interestingly, despite the broad extracellular *Ollas-Dpp* accumulation, the pMad signal dropped to background levels at the lateral regions (*Figure 2E and F*), indicating that Dpp signal activation by Dally is inefficient likely due to sequestration of Dpp by over-expressed Dally. In contrast, expression of *dlp* using *ap*-Gal4 increased HS chain levels but rather reduced extracellular *Ollas-Dpp* distribution and pMad gradient in the dorsal compartment compared to the control ventral compartment (*Figure 2C, D, G and H*), likely because ectopic Dlp expression sequestered HS (*Figure 2C*) and blocked HS modification of Dally. Consistent with a previous report (*Han et al., 2005*), Dlp expression in the dorsal compartment efficiently expanded extracellular Wg distribution compared to Dally expression (*Figure 2—figure supplement 2*). These results suggest that, although both Dally and Dlp are modified by HS chains, only Dally interacts with Dpp.

### Dally, but not Dlp, is required for the Dpp activity gradient

To compare the requirement of Dally and Dlp for Dpp signaling gradient formation, we generated null mutants for *dally* and *dlp* (*dally^{KO}* and *dlp^{KO}* respectively) by replacing the first exon of each gene by an attP cassette via genome engineering (*Figure 3—figure supplement 1*). Although there is no

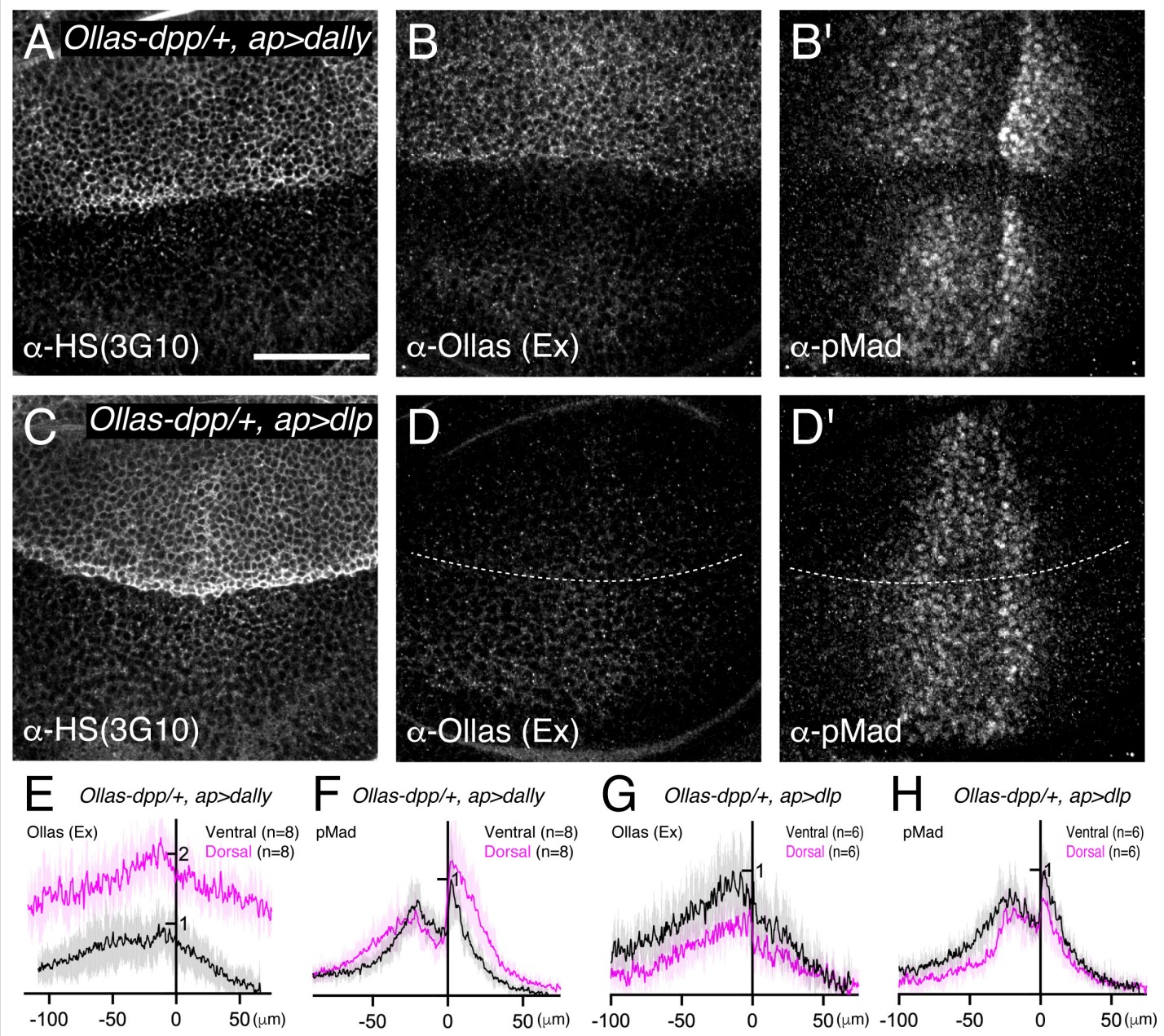

**Figure 2.** Dally, but not Dlp, interacts with Dpp. (**A–D**) α-HS (3G10) (**A, C**), extracellular α-Ollas (**B, D**), and α-pMad (**B', D'**) staining of *Ollas-dpp/+, ap >dally* disc (**A–B**) and *Ollas-dpp/+, ap >dlp* disc (**C–D**). (**E–H**) Average fluorescence intensity profile of (**B, B', D, D'**) respectively. Data are presented as mean +/- SD. Scale bar: 50 μm.

The online version of this article includes the following figure supplement(s) for figure 2:

**Figure supplement 1.** Comparison of endogenous Dpp gradient and signaling between dorsal and ventral compartment.

**Figure supplement 2.** Preferential interaction of Dlp with Wg.

good anti-Dally antibody, defects in pMad signal in *dally^{KO}* wing discs were similar to that of *dally^{MH32}* (a *dally* null allele **Franch-Marro et al., 2005** ) wing discs (**Figure 3A-C, I** ) and *dally^{KO}* adult wings showed similar phenotypes to *dally^{MH32}* adult wings (**Figure 3E–G and J**). Immunostaining for Dlp revealed that Dlp expression is completely lost in *dlp^{KO}* wing disc (**Figure 3—figure supplement 2**). Thus, we confirmed that both *dally^{KO}* and *dlp^{KO}* are null alleles. These null alleles also serve as a platform to insert epitope tags or modified versions of *dally* and *dlp* in the endogenous genomic locus via the attP cassette.

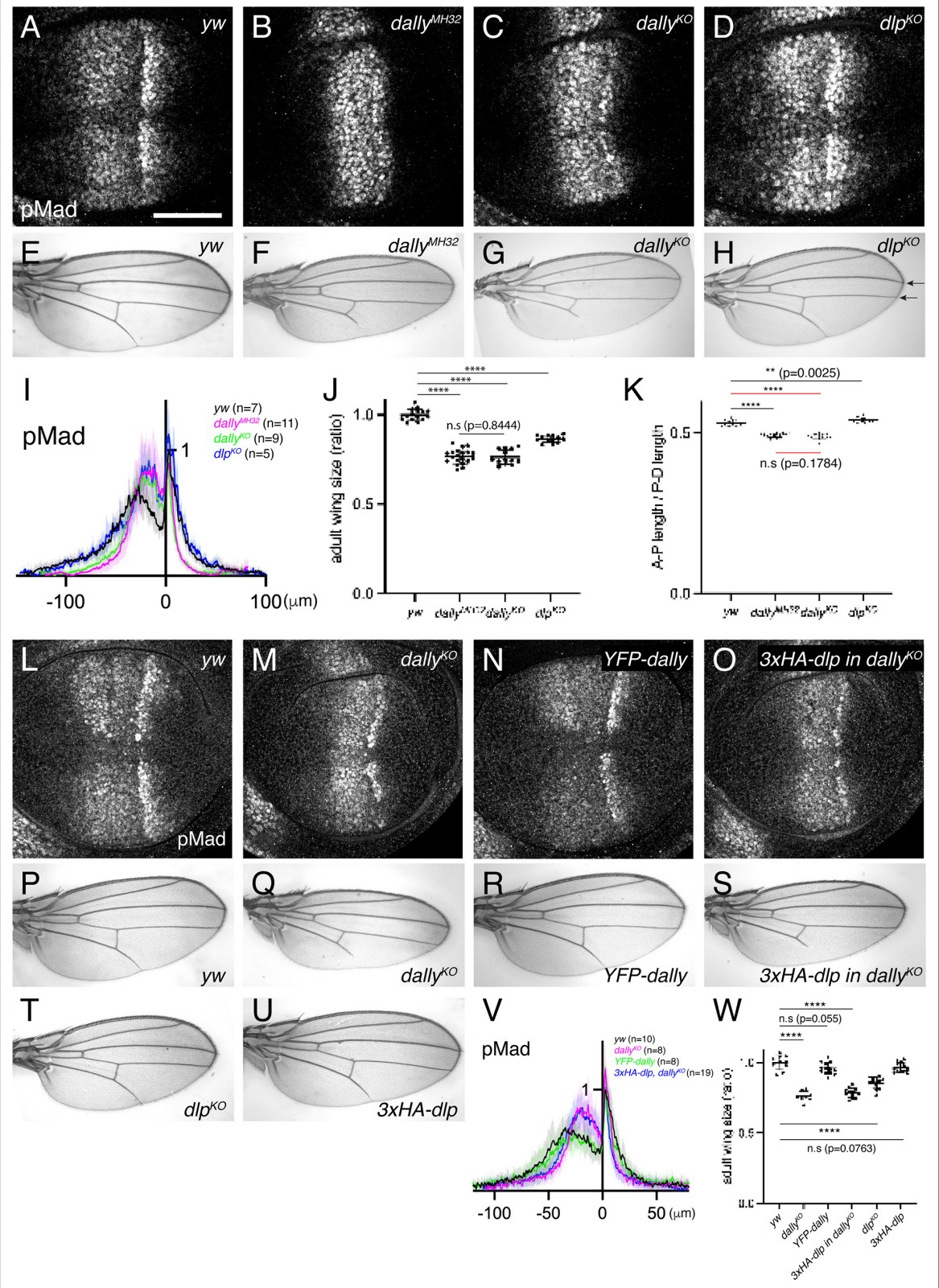

**Figure 3.** Dally, but not Dlp, is required for Dpp signaling gradient. (**A–D**) α-pMad staining of *yw* (**A**), *dally^{MH32}* (**B**), *dally^{KO}* (**C**), and *dlp^{KO}* (**D**) wing disc. (**E–H**) Adult wings of *yw* (**E**), *dally^{MH32}* (**F**), *dally^{KO}* (**G**), and *dlp^{KO}* (**H**). (**I**) Average fluorescence intensity profile of (**A–D**). Data are presented as mean +/- SD. (**J**) Comparison of adult wing size of *yw* (n=15), *dally^{MH32}* (n=20), *dally^{KO}* (n=13), and *dlp^{KO}* (n=13). Data are presented as mean +/- SD. Two-sided unpaired Student's t test with unequal variance was used for all the comparison. ****p < 0.0001. n.s; not significant. (**K**) Comparison of normalized A-P

*Figure 3 continued on next page*

*Figure 3 continued*

length against D-V length of adult wings of *yw* (n=15), *dally^MH32^* (n=20), *dally^KO^* (n=13), and *dlp^KO^* (n=13). Data are presented as mean +/- SD. Two-sided unpaired Student's t test with unequal variance was used for comparison under black lines. Two-sided Mann–Whitney test was used for comparison under red lines. ****p < 0.0001, **p < 0.01. (**L–O**) α-pMad staining of *yw* (**L**), *dally^KO^* (**M**), *YFP-dally* (**N**), and *3xHA-dlp* in *dally^KO^* (**O**) wing disc. (**P–U**) Adult wings of *yw* (**P**), *dally^KO^* (**Q**), *YFP-dally* (**R**), *3xHA-dlp* in *dally^KO^* (**S**), *dlp^KO^* (**T**), and *3xHA-dlp* (**U**). (**V**) Average fluorescence intensity profile of (**L–O**). Data are presented as mean +/- SD. (**W**) Comparison of adult wing size of *yw* (n=10), *dally^KO^* (n=8), *YFP-dally* (n=13), *3xHA-dlp* in *dally^KO^* (n=13), *dlp^KO^* (n=13), and *3xHA-dlp* (n=13). Data are presented as mean +/- SD. Two-sided unpaired Student's t test with unequal variance was used for all the comparison. ****p < 0.0001. Scale bar: 50 μm.

The online version of this article includes the following figure supplement(s) for figure 3:

**Figure supplement 1.** Generation of *dally^KO^* and *dlp^KO^* (**A**) Schematic view of manipulation of the glypican genomic loci exemplified with *dlp*.

**Figure supplement 2.** Validation of *dlp^KO^* allele.

We then compared the two null alleles and found that, while severely shrunk in *dally^KO^* wing discs, the pMad signal was not affected in *dlp^KO^* wing discs (***Figure 3C, D, I***). In rare-occurring mutant adult wings, patterning defects linked to defects in Dpp signaling (truncated L5) were only seen in *dally^KO^* mutants (***Figure 3G***). In contrast, the distance of the distal tips of L3 and L4 veins was reduced in *dlp^KO^* mutants, which is indicative of Hh signaling defects (see arrows in ***Figure 3H***). Although both *dally^KO^* and *dlp^KO^* adult wings are smaller than controls (***Figure 3J***), only *dally^KO^* adult wings are narrower along the A-P axis, consistent with the role of Dpp along this axis (***Figure 3K***).

It has been shown that *dally* and *dlp* display distinct expression patterns in the wing disc, since *dally* expression is controlled by Dpp signaling (***Fujise et al., 2003***) and *dlp* expression is controlled by Wg signaling (***Han et al., 2005***). Thus, the differential expression pattern of Dally and Dlp may underlie the differential requirements of glypicans in Dpp signaling. To address this, we inserted *dlp* in the *dally* genomic locus using the attP site in the *dally^KO^* allele, thus expressing *dlp* in the *dally* expression pattern. As a control, *YFP-dally* inserted in *dally^KO^* rescued pMad defects (***Figure 3L–N, V***) and adult wing defects of *dally^KO^* mutants (***Figure 3P–R and W***). In contrast, *3xHA-dlp* insertion into *dally^KO^* failed to rescue pMad defects (***Figure 3M, O, V***) or adult wing defects in *dally^KO^* mutants (***Figure 3Q, S and W***), although *3xHA-dlp* insertion into *dlp^KO^* restored adult wing growth in *dlp^KO^* mutants (***Figure 3T, U and W***). Taken together, these results indicate that *dally*, but not *dlp*, is required for Dpp signaling gradient formation.

## Interaction of the core protein of Dally with Dpp

How can only Dally be involved in Dpp signaling although both HSPGs are modified with HS chains (***Figure 2A and C***)? Dally has been shown to interact with Dpp in vitro not only through the HS chains but also through the core protein (***Kirkpatrick et al., 2006***). Thus, we wondered whether the core protein of Dally can interact with Dpp in the wing disc to provide ligand specificity. To test this, we expressed *dally* or *dally^ΔHS^*, a *dally* mutant lacking all HS chain attachment sites (***Kirkpatrick et al., 2006***) using *ap*-Gal4 to compare their ability to interact with Dpp and activate pMad signal. Upon *dally* expression in the dorsal compartment, HS chains, extracellular Dpp, and pMad signal increased as compared to the control, ventral compartment (***Figure 4A–C, G and J***). In contrast, upon *dally^ΔHS^* expression, HS chains rather decreased (***Figure 4D***), probably because the core protein of Dally sequesters the factors required for synthesis of HS chains. Under this condition, extracellular Dpp still increased but pMad signal rather decreased (***Figure 4E, F, H and K***). These results indicate that the core protein of Dally can interact with Dpp but this interaction is not sufficient and requires the HS chains to activate pMad signal.

Since Dally^ΔHS^ expression was higher than Dally expression (***Figure 4B' and E'***), we measured the relative Dpp accumulation against Dally or Dally^ΔHS^ expression (***Figure 4I***) to address the relative contribution of the core protein and the HS chains for Dpp binding. We found that the relative Dpp accumulation upon *dally^ΔHS^* expression is only slightly reduced compared to that upon *dally* expression (***Figure 4I***), indicating a major contribution of the core protein in the interaction between Dally and Dpp in the wing disc.

## HS chains of Dally are critical for Dpp distribution and signaling

The relatively minor contribution of HS chains of Dally for the interaction with Dpp raises questions on their requirement in Dpp distribution and signaling under physiological conditions. To address this, we

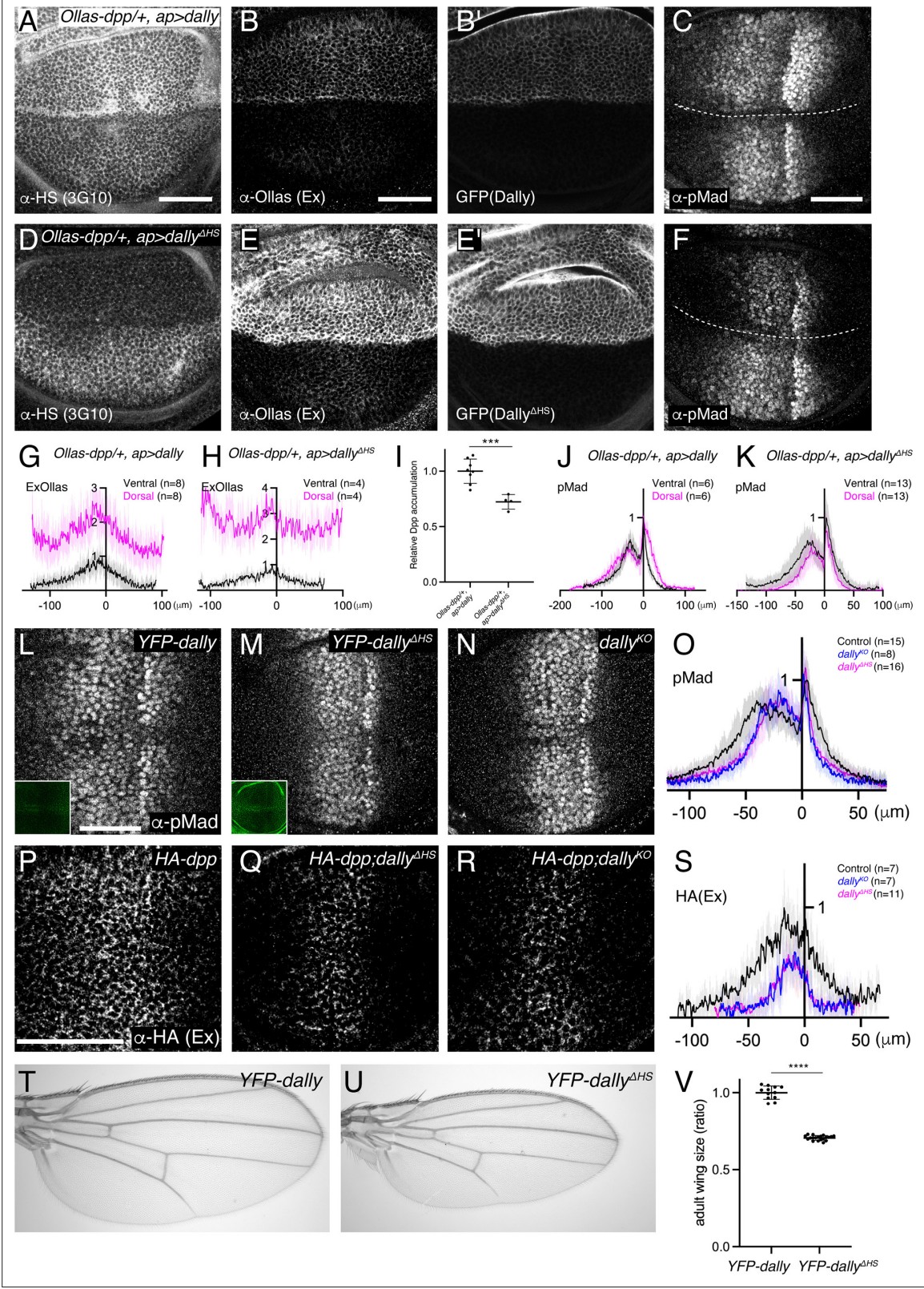

**Figure 4.** Interaction of core protein of Dally with Dpp. (**A–F**) α-HS (3G10) (**A, D**), extracellular α-Ollas (**B, E**), GFP (**B', E'**), α-pMad (**C, F**) staining of *Ollas-dpp/+, ap >dally* disc (**A–C**) and *Ollas-dpp/+, ap >dally^ΔHS* disc (**D–F**). (**G**) Average fluorescence intensity profile of (**B**). (**H**) Average fluorescence intensity profile of (**E**). (**I**) Relative Dpp accumulation (α-Ollas signal) (**B, E**) normalized against expression level of Dally or Dally^ΔHS (GFP fluorescent signal) around the Dpp producing cells (**B', E'**). Two-sided unpaired Student's t test with unequal variance was used for the comparison. ***p< 0.001.

*Figure 4 continued on next page*

Figure 4 continued

(**J**) Average fluorescence intensity profile of (**C**). (**K**) Average fluorescence intensity profile of (**F**). (**G–K**) Data are presented as mean +/- SD. (**L–N**) α-pMad staining of YFP-dally (**L**), YFP-dally$^{\Delta HS}$ (**M**), and dally$^{KO}$ (**N**) wing disc. YFP signal of YFP-dally (**L, inset**) and YFP-dally$^{\Delta HS}$ (**M, inset**) wing disc. We note that, similar to the overexpression results (**B' and E'**), YFP-dally$^{\Delta HS}$ wing discs showed higher expression levels than YFP-dally$^{\Delta HS}$ (**L and M inset**). (**O**) Average fluorescence intensity profile of (**L–N**). Data are presented as mean +/- SD. (**P–R**) Extracellular α-HA staining of HA-dpp (**P**), HA-dpp; YFP-dally$^{\Delta HS}$ (**Q**), and HA-dpp; dally$^{KO}$ (**R**) wing disc. (**S**) Average fluorescence intensity profile of (**P–R**). Data are presented as mean +/- SD. (**T–U**) Adult wings of YFP-dally (**T**) and YFP-dally$^{\Delta HS}$ (**U**). (**V**) Comparison of adult wing size of YFP-dally (n=12) and YFP-dally$^{\Delta HS}$ (n=20). Data are presented as mean +/- SD. Two-sided unpaired Student's t test with unequal variance was used for the comparison. ****p < 0.0001. Scale bar: 50 µm.

inserted YFP-dally$^{\Delta HS}$, a dally mutant allele lacking HS chain modification (**Kirkpatrick et al., 2006**) in the attP site of the dally$^{KO}$ allele. We found that both pMad signaling (**Figure 4L–O**) and extracellular Dpp distribution (**Figure 4P-S**) were severely affected in the YFP-dally$^{\Delta HS}$ wing discs, thus resembling the phenotypes of dally$^{KO}$ mutants. Consistently, the YFP-dally$^{\Delta HS}$ adult wings mimicked patterning and growth defects of dally$^{KO}$ adult wings (**Figure 4T–V**). These results show that the interaction of the core protein of Dally with Dpp is not sufficient and that the HS chains of Dally are critical for Dpp distribution and signaling under physiological conditions.

## HS chains of Dally control Dpp distribution and signaling largely independent of interaction with Dpp

How do HS chains of Dally control Dpp distribution and signaling gradient? The interaction of the HS chains of glypicans with Dpp is thought to be essential for their function. However, our data suggest that, while important for proper Dpp distribution and signaling, HS chains have a minor contribution to the binding of Dpp. To address whether a direct interaction of the HS chains of Dally and Dpp is critical, we sought to generate a Dpp variant that lacks HS-binding properties. A previous study showed that seven basic amino acids at the N-terminus of the Dpp mature domain are crucial for the interaction with heparin but dispensable for interaction with receptors to activate downstream signaling (**Akiyama et al., 2008**). To test the importance of this interaction in vivo, we generated dpp$^{\Delta N}$, a dpp mutant allele lacking the basic amino acid stretch. We found that the dpp$^{\Delta N}$ allele is embryonic lethal, likely because the basic amino acids overlap with a collagen binding site and the Dpp-Collagen IV interaction is important for Dpp gradient formation in the early embryo (**Wang et al., 2008**). To bypass this embryonic lethality without affecting wing disc development, we utilized a previously reported transgene called JAX, which contains the genomic region of dpp (JAX) critical for the early embryogenesis (**Hoffmann and Goodman, 1987**) but does not rescue the wing phenotypes of dpp mutants (**Figure 5—figure supplement 1**). Indeed, the early lethality of homozygous HA-dpp$^{\Delta N}$ was greatly rescued in the presence of this transgene (JAX; HA-dpp$^{\Delta N}$), thus allowing to investigate the requirement of the interaction of Dpp with the HS chains of Dally in later stages.

To test if Dpp$^{\Delta N}$ lacks interaction with HS chains of Dally, we expressed dally or dally$^{\Delta HS}$ in JAX;dpp$^{\Delta N}$ using ap-Gal4 and compared their ability to interact with Dpp$^{\Delta N}$ (**Figure 5A and B**). We found that, in both cases, extracellular distribution of Dpp$^{\Delta N}$ increased in the dorsal compartment (**Figure 5A and B**), indicating that Dpp$^{\Delta N}$ can interact with the core protein of Dally. To address the relative contribution of the HS chains and the core protein for Dpp$^{\Delta N}$-Dally interaction, Dpp$^{\Delta N}$ accumulation was normalized against Dally or Dally$^{\Delta HS}$ expression since Dally$^{\Delta HS}$ expression was higher than Dally expression (**Figure 5A' and B'**). We found that the resulting normalized Dpp$^{\Delta N}$ accumulation was not changed between two conditions (**Figure 5C**), indicating that the observed Dpp$^{\Delta N}$ accumulation upon Dally expression is mainly through Dally core protein and Dpp$^{\Delta N}$ indeed lacks interaction with HS chains.

Using the same experimental setup, we then compared the ability of Dally and Dally$^{\Delta HS}$ to activate Dpp signal in JAX;dpp$^{\Delta N}$ (**Figure 5A" and B"**), in which HS chains-Dpp interaction is lost (**Figure 5C**). We found that Dally expression (**Figure 5A" and D**), but not Dally$^{\Delta HS}$ expression (**Figure 5B" and E**), expanded Dpp signaling in JAX;dpp$^{\Delta N}$, indicating that Dally can activate Dpp signaling through HS chains but independent of HS chains-Dpp interaction.

Surprisingly, and in contrast to dally$^{KO}$ or dally$^{\Delta HS}$ flies (**Figures 3 and 4**), we found that JAX; HA-dpp$^{\Delta N}$ allele showed very subtle defects in Dpp distribution and signaling in wing discs (**Figure 5F–I**) and in patterning and growth in adult wings (**Figure 5J–L**). Indeed, while dally$^{KO}$ and dally$^{\Delta HS}$ adult wings are 25–30% smaller than control wings (**Figures 3J, W and 4V**), JAX; HA-dpp$^{\Delta N}$ adult wings are only 6.4% smaller than control wings (**Figure 5L**). Taken together, these results suggest that, although the HS

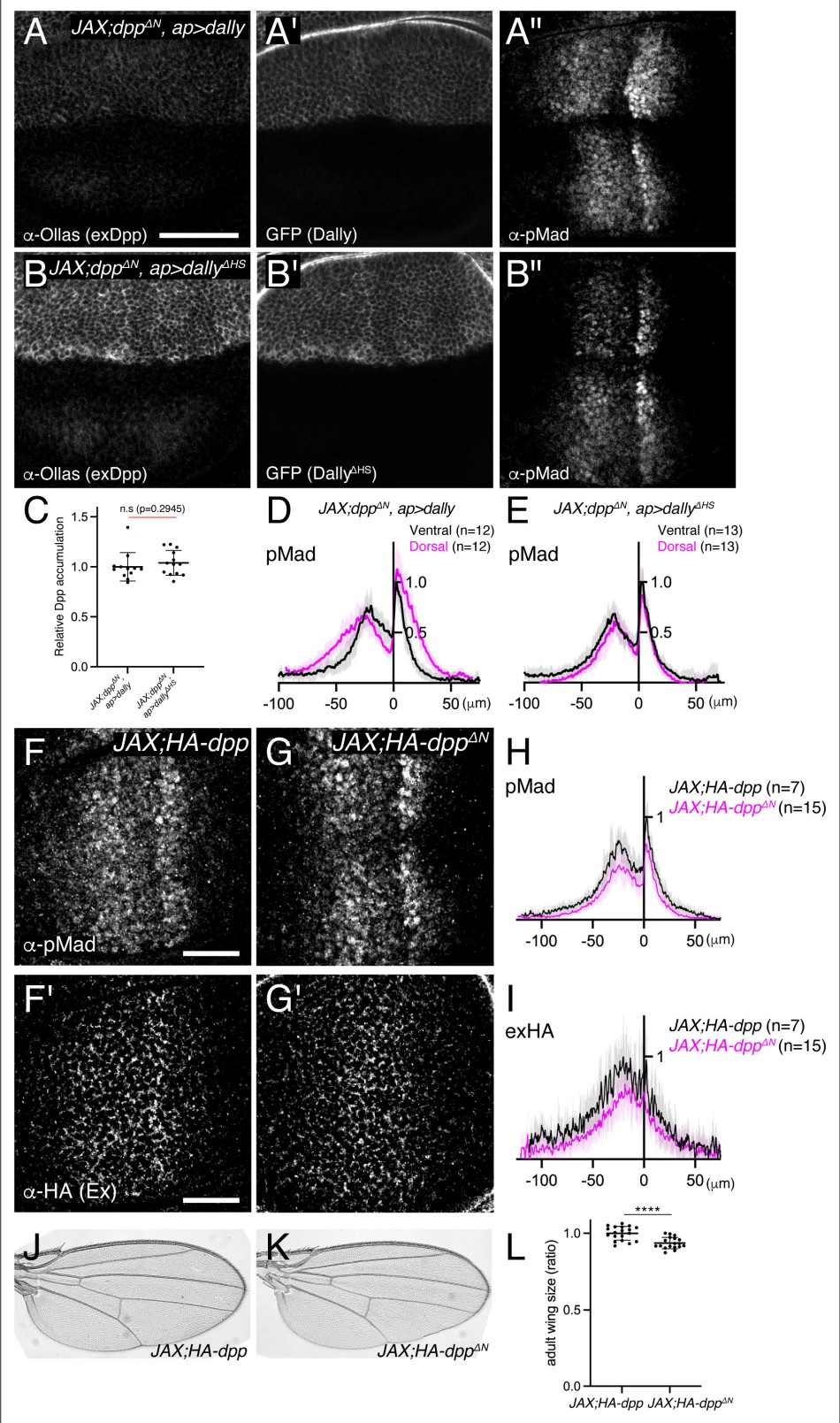

**Figure 5.** HS chains of Dally act largely independent of interaction with Dpp. (**A–B**) extracellular α-Ollas (**A, B**), GFP (**A', B'**), and α-pMad (**A", B"**) staining of $JAX;HA-dpp^{ΔN}$, $ap > dally$ disc (**A-A"**) and $JAX;HA-dpp^{ΔN}$, $ap > dally^{ΔHS}$ disc (**B-B"**). (**C**) Relative Dpp accumulation (α-Ollas signal) (**A, B**) normalized against expression level of Dally or Dally$^{ΔHS}$ (GFP fluorescent signal) around the Dpp producing cells (**A', B'**). Two-sided Mann–Whitney

*Figure 5 continued on next page*

*Figure 5 continued*

test was used for comparison. n.s; not significant. (**D**) Average fluorescence intensity profile of (**A″**). Data are presented as mean +/- SD. (**E**) Average fluorescence intensity profile of (**B″**). Data are presented as mean +/- SD. (**F–G**) α-pMad (**F, G**) and extracellular α-HA (**F′, G′**) staining of *JAX;HA-dpp* disc (**F, F′**) and *JAX;HA-dpp^{ΔN}* disc (**G, G′**). (**H**) Average fluorescence intensity profile of (**F, G**). Data are presented as mean +/- SD. (**I**) Average fluorescence intensity profile of (**F′, G′**). Data are presented as mean +/- SD. (**J–K**) Adult wings of *JAX;HA-dpp* (**J**) and *JAX;HA-dpp^{ΔN}* (**K**). (**L**) Comparison of adult wing size of *JAX;HA-dpp* (n=19) and *JAX;HA-dpp^{ΔN}* (n=18). Data are presented as mean +/- SD. Two-sided unpaired Student's t test with unequal variance was used for the comparison. ****p < 0.0001.

The online version of this article includes the following figure supplement(s) for figure 5:

**Figure supplement 1.** *JAX* does not rescue *dpp* disc allele.

chains of Dally are essential for Dpp distribution and signaling (*Figure 4*), a direct interaction of HS chains of Dally with Dpp is largely dispensable (*Figure 5*).

## HS chains of Dally stabilize Dpp by antagonizing Tkv-mediated Dpp internalization

So far, all models on how glypicans control Dpp distribution and signaling are based on direct inter-action of the HS chains with Dpp (*Figure 1*). How can the HS chains of Dally control Dpp distribution and signaling without directly interacting with Dpp? Among the models, we wondered whether Tkv is a factor through which HS chains of Dally can act (*Figure 1D*), although HS chains are thought to act through Dpp in the original model (*Akiyama et al., 2008*).

We first revisited the role of Tkv. According to the model (*Akiyama et al., 2008*; *Figure 1D*), Tkv is the Dpp receptor on the cell surface that internalizes and degrades Dpp in addition to activating pMad signal and Dally antagonizes Tkv-mediated internalization of Dpp to control extracellular Dpp distribution. However, a recent study proposed that Dally, but not Tkv, internalizes Dpp (*Romanova-Michaelides et al., 2022*). To address this discrepancy, we first depleted *tkv* by RNAi using *ap*-Gal4 and visualized endogenous extracellular Dpp distribution. If Tkv is required for Dpp internalization, extracellular Dpp distribution should increase by removing Tkv. Indeed, we found that extracellular Dpp increased and expanded in the dorsal compartment (*Figure 6A and C*), indicating that Tkv is required to internalize Dpp.

To address whether Dally antagonizes Tkv-mediated Dpp internalization, we asked if the defects in Dpp distribution in *dally* mutants can be rescued by RNAi-mediated depletion of *tkv* using *ap*-Gal4. We found that the extracellular Dpp gradient expanded in the dorsal compartment (*Figure 6B and D*), indicating that Dally antagonizes Tkv-mediated Dpp internalization. Expanded Dpp distribution in *dally* mutants upon blocking Dpp internalization also suggests that Dally is largely dispensable for Dpp spreading, and that there are factors other than Tkv and Dally that bind to Dpp on the cell surface.

We then tested if Dally can promote Dpp internalization independent of antagonizing Tkv-mediated Dpp internalization. In this scenario, extracellular Dpp distribution in the absence of *tkv* should further increase by removing *dally*. To test this, we compared extracellular Dpp levels in the dorsal compart-ment of *ap >tkv* RNAi wing discs (*Figure 6A*) and that of *ap >tkv RNAi, dally^{KO}* wing discs (*Figure 6B*). However, we found that extracellular Dpp levels do not further increase (*Figure 6E*), indicating that Dally is not essential to internalize Dpp.

Interestingly, when *tkv* was depleted in the dorsal compartment, pMad was not only lost from the dorsal compartment but also upregulated in ventral cells along the dorsal-ventral compartment boundary (arrow, *Figure 6F*). This observation is consistent with the idea that, upon blocking inter-nalization, Dpp from the dorsal compartment can reach to the ventral compartment to activate pMad through Tkv binding. In contrast, this phenotype was not seen upon depletion of *dally* by RNAi, although depletion of *dally* in the dorsal compartment mimicked the pMad defects of *dally* mutants (*Figure 6G*).

While HS chains of Dally are thought to compete with Tkv for binding to Dpp in the original model (*Akiyama et al., 2008*; *Figure 1D*), the minor phenotypes of *dpp^{ΔN}* (*Figures 4 and 5*) suggest that HS chains act through a different mechanism. To test if the HS chains of Dally stabilizes Dpp on the cell surface by antagonizing Tkv-mediated Dpp internalization, we depleted *tkv* by RNAi using *ap*-Gal4

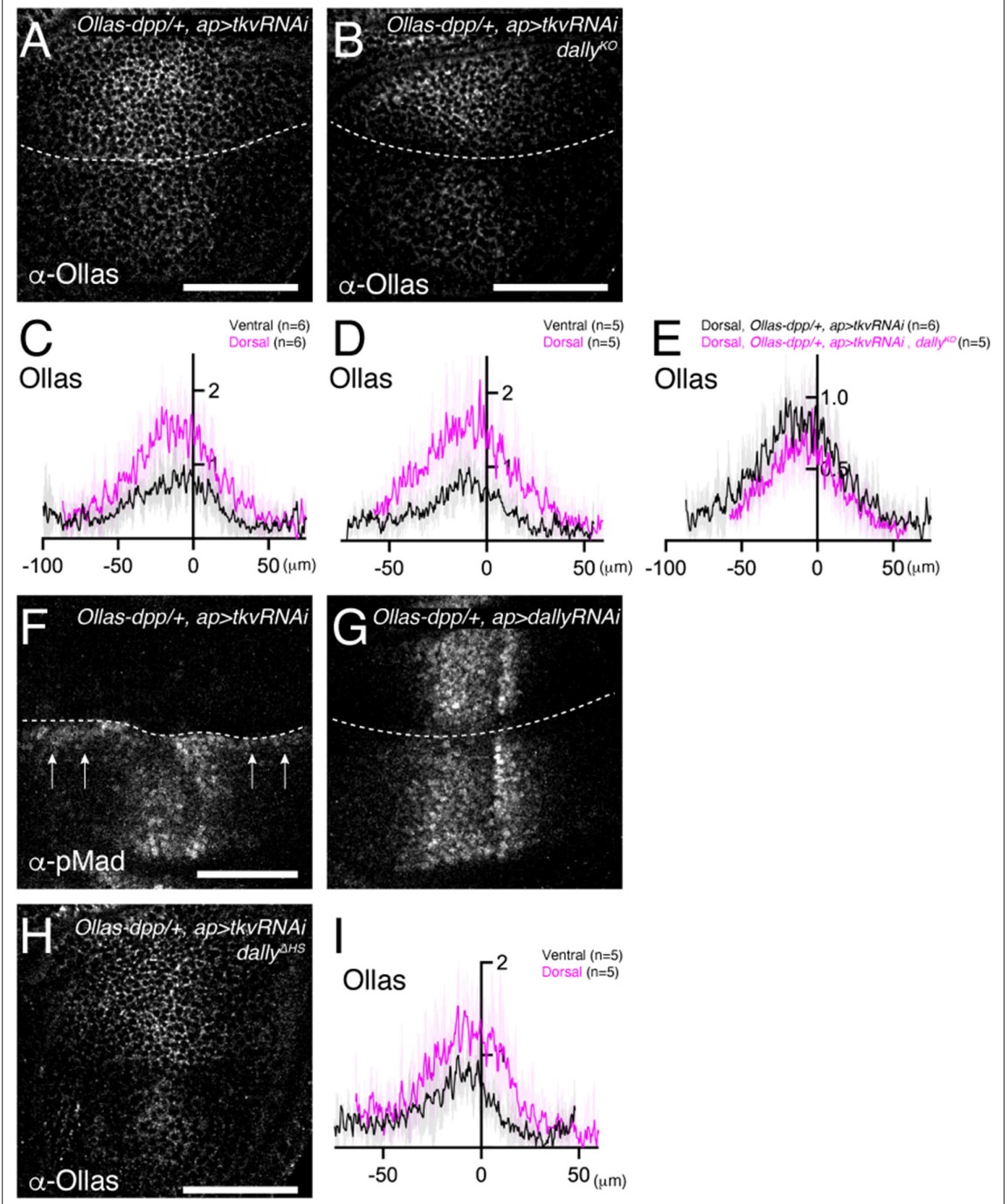

**Figure 6.** HS chains of Dally stabilize Dpp by antagonizing Tkv-mediated Dpp internalization. (**A, B**) Extracellular α-Ollas staining of *Ollas-dpp/+, ap >tkv* RNAi wing disc (**A**) and *dally^KO^, Ollas-dpp/+, ap >tkv* RNAi wing disc (**B**). (**C, D**) Average fluorescence intensity profile of (**A, B**) respectively. Data are presented as mean +/- SD. (**E**) Average fluorescence intensity profile of the dorsal compartment of (**A, B**). Data are presented as mean +/- SD. (**F, G**) α-pMad staining of *Ollas-dpp/+, ap >tkv* RNAi wing disc (**F**), and *Ollas-dpp/+, ap >dally* RNAi wing disc (**G**). (**H**) Extracellular α-Ollas staining of *dally^ΔHS^, Ollas-dpp/+, ap >tkv* RNAi wing disc. (**I**) Average fluorescence intensity profile of (**H**). Data are presented as mean +/- SD. Scale bar: 50 μm.

in *dally^ΔHS^* mutants. We found that extracellular Dpp gradient expanded in the dorsal compartment (***Figure 6H,I***), thus indicating that the HS chains of Dally antagonize Tkv-mediated Dpp internalization.

Cumulatively, these results suggest that Tkv, but not Dally, is required to internalize Dpp, and support a model, in which HS chains of Dally stabilizes Dpp on the cell surface by blocking Tkv-mediated Dpp internalization (*Akiyama et al., 2008*).

## Discussion

HSPGs are essential for Dpp/BMP morphogen gradient formation and signaling in the *Drosophila* wing disc. Although a variety of models have been proposed, it remains largely unknown how distinct HSPGs control Dpp gradient formation and signaling. In this study, we generated and utilized genome engineering platforms for two HSPGs, *dally* and *dlp*, and *dpp* (*Matsuda et al., 2021*) to address this question.

### Mechanism of Dally function

It has been thought that Dally and Dlp act redundantly to control Dpp signaling. However, we demonstrate here that Dally, but not Dlp, is critical for Dpp distribution and signaling through interaction of its core protein with Dpp (*Figures 2–4*). We do not know if the interaction is direct but previous surface plasmon resonance (SPR) and co-immunoprecipitation experiments suggest that this is the case (*Kirkpatrick et al., 2006*). It has recently been shown that a specific structure, present in the core protein of Dlp but absent in Dally, shields the lipid moiety of Wg and is important for its spreading (*McGough et al., 2020*). Thus, each glypican appears to acquire ligand specificity through its core protein.

In contrast to Wg, Dpp has no lipid modification and is diffusible by nature. Indeed, the expanded Dpp distribution in *dally* mutants upon blocking Dpp internalization (*Figure 6*) does not support the idea that Dally is required for Dpp spreading itself (*Belenkaya et al., 2004*; *Stapornwongkul et al.,*

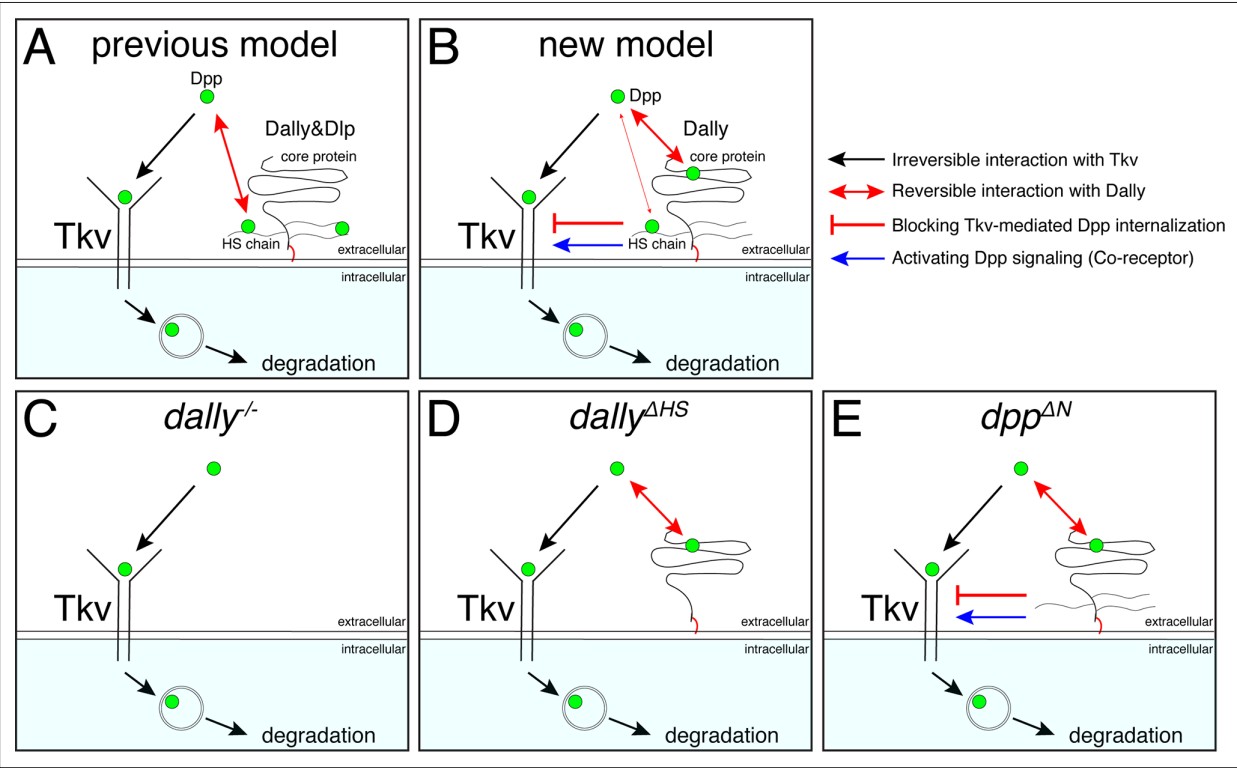

**Figure 7.** A revised model on the roles of Dally on Dpp/BMP gradient formation and signaling. (**A**) A previous model. Dally and Dlp competes with Tkv for Dpp binding to antagonize Tkv-mediated Dpp internalization. (**B**) A revised model. Dally, but not Dlp, reversibly interacts with Dpp partially through its HS chains and mainly through its core protein. The former interaction is not essential and the latter interaction is not sufficient to antagonize Tkv-mediated Dpp internalization under physiological conditions. Upon interaction of Dpp with the core protein of Dally, HS chains of Dally antagonize Tkv-mediated Dpp internalization through Tkv to stabilize Dpp on the cell surface. HS chains of Dally control Dpp signaling also independent of interaction with Dpp likely through Tkv. It remains unknown whether and how Dpp signal activation by core protein bound Dpp and stability by HS chains are coordinated. (**C**) In *dally^{KO}* mutants, Tkv-mediated Dpp internalization by Tkv is not antagonized. Dpp is irreversibly bound to Tkv and then removed by Tkv from the extracellular space and degraded. (**D**) In *dally^{ΔHS}* mutants, the core protein of Dally can interact with Dpp but this interaction is not sufficient to antagonize Tkv-mediated Dpp internalization. Without HS chains, Dpp is irreversibly bound to Tkv and then removed by Tkv from the extracellular space and degraded. (**E**) In *dpp^{ΔN}* mutants, Dpp can interact with core protein but not with HS chains of Dally. Without interaction of HS chains with Dpp, HS chains can antagonize Tkv-mediated Dpp internalization through Tkv to stabilize Dpp on the cell surface.

*2020*; *Schwank et al., 2011*; *Figure 1B*) or that Dally and Tkv are essential for blocking Dpp leakage from the wing epithelia (*Stapornwongkul et al., 2020*). This result suggests that there are other cell surface molecules that bind to Dpp on the cell surface. It remains to be tested if Dpp spreads by free diffusion or requires such Dpp binding proteins. It is also unlikely that Dally inhibits Dpp spreading (*Figure 1C*), since extracellular Dpp distribution was not blocked upon *dally* expression (*Figure 2*) or expanded in *dally* mutants (*Figures 4R and 6E*). Our results thus indicate that Dally acts downstream of Dpp spreading.

Despite the interaction of the core protein of Dally with Dpp upon Dally overexpression, this interaction is not sufficient to control Dpp distribution and Dpp signaling at its physiological level, and HS chains of Dally are absolutely essential for both processes (*Figure 4*). Since increase of HS chains was not sufficient for interaction with Dpp (*Figure 2C and D*), we speculate that HS chains of Dally act downstream of the interaction of the core protein of Dally with Dpp.

Among the proposed models (*Figure 1*), our results support a model, in which Dally stabilizes Dpp on the cell surface by antagonizing Tkv-mediated internalization of Dpp (*Akiyama et al., 2008*; *Figures 1D and 7A*). Consistently, pulse-chase experiments showed that Dpp is less stable in *dally* mutants (*Akiyama et al., 2008*). However, while the original model indicates that HS chains compete with Tkv for Dpp binding (*Akiyama et al., 2008*; *Figures 1D and 7A*), our results suggest that HS chains act largely through Tkv based on two observations (*Figure 7B*). First, although the *dally*^ΔHS allele showed severe defects similar to the *dally*^KO allele, the defects of the *dpp*^ΔN allele were surprisingly mild despite the lack of interaction with HS chains (*Figures 4 and 5*), indicating that HS chains act largely through other molecules. Second, defects in Dpp distribution in *dally*^ΔHS were rescued by removing *tkv* (*Figure 6*). These results indicate that HS chains act through Tkv, rather than through Dpp, to stabilize Dpp on the cell surface (*Figure 7C–E*).

It remains unknown how HS chains of Dally stabilize Dpp through Tkv. Since heparin has been shown to bind to BMP receptors (*Kanzaki et al., 2008*), the HS chains of Dally may directly interact with Tkv to stabilize Dpp on the cell surface. Interestingly, we found that the HS chains of Dally control Dpp signaling also largely independent of interaction with Dpp (*Figure 5D and E*), which likely reflects the role of Dally as a co-receptor. The simplest possibility is that the HS chains of Dally control both Dpp stability and signaling through Tkv (*Figure 7*), but it is equally possible that the HS chains of Dally control Dpp signaling through other factors. In the future, it will be also important to address how the interaction of the core protein of Dally with Dpp and functions of HS chains through Tkv are coordinated.

## Mechanisms of Dpp internalization

Our proposed model is based on the result that Dpp is internalized by Tkv (*Figure 6*), consistent with a widely accepted idea that extracellular morphogens are removed by their receptors. However, this result contradicts a recent report showing that Tkv is not required to internalize Dpp (*Romanova-Michaelides et al., 2022*). How can we explain the discrepancy?

If Tkv is involved in internalization of Dpp, internalized Dpp should decrease upon loss of *tkv*. Romanova-Michaelides et al. applied a nanobody internalization assay to visualize internalized Dpp (*Romanova-Michaelides et al., 2022*). In this assay, extracellular GFP-Dpp was first labeled using fluorophore conjugated anti-GFP nanobodies under endocytosis-restricted condition. After allowing internalization of the labelled GFP-Dpp, extracellular anti-GFP nanobodies were then stripped away by acid wash to follow only the internalized Dpp. Using this assay, Romanova-Michaelides et al. surprisingly found that internalization of GFP-Dpp was not affected in *tkv* mutant clones. However, the experiment was performed under overexpression of GFP-Dpp from the anterior stripe of cells, in which the authors estimate 400 times higher expression than the physiological level. Under such unphysiological conditions, Dpp internalization through other Dpp binding factors can easily mask the impact of Tkv on Dpp internalization. Consistently, we found that Dpp binds to the cell surface without Tkv and Dally (*Figure 6*).

If Tkv is involved in Dpp internalization, extracellular Dpp should increase upon loss of *tkv*. Romanova-Michaelides et al. used the observation that extracellular GFP-Dpp did not increase in *tkv* mutant clones upon GFP-Dpp expression from the anterior stripe of cells (*Schwank et al., 2011*) to further support Tkv-independent Dpp internalization (*Romanova-Michaelides et al., 2022*). However, under 400 times higher expression levels of GFP-Dpp than the physiological level, it is not surprising

that an increase of extracellular Dpp can be missed. Indeed, we find that endogenous extracellular Dpp indeed increased upon *tkv* depletion, indicating that Dpp is internalized through Tkv (*Figure 6*). Notably, even under GFP-Dpp overexpression, accumulation of GFP-Dpp on the surfaces of cells lacking Tkv have been previously reported (*Entchev et al., 2000*).

Using the nanobody internalization assay, Romanova-Michaelides et al. showed that Dally, but not Tkv, is required for internalization of Dpp based on reduced internalization of endogenous GFP-Dpp upon blocking HS chain synthesis or upon cleaving HSPGs (*Romanova-Michaelides et al., 2022*). In contrast, our results suggest that Dally rather antagonizes internalization of Dpp by blocking Tkv-mediated internalization of Dpp (*Figure 6*). How can Dpp internalization be reduced in *dally* mutants if Dally antagonizes internalization of Dpp? We found that extracellular Dpp distribution is severely reduced in *dally* mutants due to Tkv-mediated internalization of Dpp (*Figure 6*), indicating that the amount of extracellular Dpp available for labeling with nanobody is reduced in *dally* mutants. Thus, we speculate that the reduced internalization of Dpp in *dally* mutants revealed by the nanobody internalization assay simply reflects the reduced extracellular Dpp rather than reduced ability to internalize Dpp.

Our results do not thus support Dally-mediated internalization and recycling of Dpp as a mechanism of Dpp gradient scaling (*Romanova-Michaelides et al., 2022*; *Figure 1E*), although recycling of Dpp through other factors may contribute to Dpp gradient scaling. It would be interesting to test if the Dally-mediated Dpp stabilization on the cell surface is involved in the Dpp gradient scaling. Consistent with this, decrease of Dpp degradation rates has previously been proposed as a mechanism for Dpp gradient scaling (*Wartlick et al., 2011*). Furthermore, previous pulse-chase experiments showed that, similar to Dally, Pentagone (Pent), a secreted feedback factor required for Dpp gradient scaling, is required for Dpp stability (*Vuilleumier et al., 2010*). Interestingly, Pent has been shown to interact with HS chains of Dally (*Vuilleumier et al., 2010*; *Norman et al., 2016*) raising a possibility that interaction of HS chains of Dally with Pent is critical for Dpp stability, thereby Dpp gradient scaling.

## Conclusion

To recognize and activate the specific signal, the affinity of a morphogen to its receptors is high. However, the high affinity of the morphogen-receptor interaction could at the same time compromise the range of morphogens through receptor-mediated removal of morphogens from the extracellular space. Thus, there must exist mechanisms ensuring long-range action of the morphogen despite its high receptor affinity. Our results suggest that Dally counteracts the internalization of Dpp through Tkv to increase the chance that Dpp can reach further despite the high affinity of Dpp to Tkv.

## Materials and methods
### Data reporting

No statistical methods were used to predetermine sample size. The experiments were not randomized, and investigators were not blinded to allocation during experiments and outcome assessment.

### Fly stocks

Flies for experiments were kept at 25 °C in standard fly vials containing polenta and yeast. The following fly lines were used: *UAS-tkv-RNAi* (Bloomington 40937), *UAS-dally-RNAi* (VDRC 14136), *Ollas-dpp* (*Bauer et al., 2023*), *UAS-GFP-HA-dally*, *UAS-GFP-dally^{ΔHS}* (Suzanne Eaton), *UAS-dlp* (*Baeg et al., 2004*), *HA-dpp* (*Matsuda et al., 2021*), *HA-dpp^{ΔN}* (this study), *dlp [ko;attB]* (this study), *dally[ko;attB]* (this study), *dally[YFP-dally;attP]* (this study), *dlp[3xHA-dlp;attP]* (this study) *dally[YFP-dally^{ΔHS};attP]* (this study), *dally[3xHA-dlp;attP]* (this study), *ap-Gal4* (Bloomington 3041), ap[c1.4b-Gal4] (Michèle Sickmann and Martin Müller, unpublished), *JAX* (*Hoffmann and Goodman, 1987*), *Ollas-HA-dpp^{ΔN}* (this study), *hh-Gal4* (Gift from Dr Manolo Calleja), *dally^{MH3237}*. *yw*, *dpp^{d8}* and *dpp^{d12}* are described from Flybase.

## Genotypes by figures

---

*Figure 2 A and B*; ap-Gal4, Ollas-dpp/+; UAS-GFP-HA-dally/+

---

*Figure 2C and D*; ap-Gal4, Ollas-dpp/+; UAS-dlp/+

---

*Figure 3A and E*; yw (control)

---

*Figure 3B and F*; dally$^{MH32}$/dally$^{MH32}$

---

*Figure 3C and G*; dally[KO;attB]/dally[KO;attB]

---

*Figure 3D and H*; dlp[KO;attB]/dlp[KO;attB]

---

*Figure 3L and P*; yw (control)

---

*Figure 3M and Q*; dally[KO;attB]/dally[KO;attB]

---

*Figure 3N and R*; dally[YFP-dally;attP]/dally[YFP-dally;attP]

---

*Figure 3O and S*; dally[3xHA-dlp;attP]/dally[3xHA-dlp;attP]

---

*Figure 3T*; dlp[KO;attB]/dlp[KO;attB]

---

*Figure 3U*; dlp[3xHA-dlp;attP]/dlp[3xHA-dlp;attP]

---

*Figure 4A–C*; ap-Gal4, Ollas-dpp/+; UAS-GFP-HA-dally/+

---

*Figure 4D–F*; ap-Gal4, Ollas-dpp/+; UAS-GFP-dally$^{\Delta HS}$

---

*Figure 4L*; dally[YFP-dally;attP]/dally[YFP-dally;attP]

---

*Figure 4M*; dally[YFP-dally$\Delta$HS;attP]/dally[YFP-dally$\Delta$HS;attP]

---

*Figure 4N*; dally[KO;attB]/dally[KO;attB]

---

*Figure 4P*; HA-dpp/HA-dpp

---

*Figure 4Q*; HA-dpp/HA-dpp;dally[YFP-dally$\Delta$HS;attP]w+/dally[YFP-dally$\Delta$HS;attP]w+

---

*Figure 4R*; HA-dpp/HA-dpp;dally[KO;attB]/dally[KO;attB]

---

*Figure 4T*; dally[YFP-dally;attP]/dally[YFP-dally;attP]

---

*Figure 4U*; dally[YFP-dally$\Delta$HS;attP]/dally[YFP-dally$\Delta$HS;attP]

---

*Figure 5A*; JAX; ap-Gal4, HA-dpp$^{\Delta N}$/ Ollas-HA-dpp$^{\Delta N}$;UAS-GFP-HA-dally/+

---

*Figure 5B*; JAX; ap-Gal4, HA-dpp$^{\Delta N}$/ Ollas-HA-dpp$^{\Delta N}$;UAS-GFP-dally$^{\Delta HS}$/+

---

*Figure 5F and J*; JAX;HA-dpp/HA-dpp

---

*Figure 5G and K*; JAX;HA-dpp$^{\Delta N}$/HA-dpp$^{\Delta N}$

---

*Figure 6A*; ap-Gal4, Ollas-dpp/UAS-tkvRNAi

---

*Figure 6B*; ap-Gal4, Ollas-dpp/UAS-tkvRNAi;dally[KO;attB]/dally[KO;attB]

---

*Figure 6F*; ap-Gal4, Ollas-dpp/UAS-tkvRNAi

---

*Figure 6G*; ap-Gal4, Ollas-dpp/+; UAS-dallyRNAi/+

---

*Figure 6H*; ap-Gal4, Ollas-dpp/UAS-tkvRNAi;dally[YFP-dally$\Delta$HS;attP]w+/dally[YFP-dally$\Delta$HS;attP]w+

---

*Continued on next page*

*Continued*

| |
| --- |
| *Figure 2—figure supplement 1A*, C; *Ollas-dpp/Ollas-dpp* |
| *Figure 2—figure supplement 2A*; *ap-Gal4, Ollas-dpp/+* |
| *Figure 2—figure supplement 2B*; *ap-Gal4, Ollas-dpp/+; UAS-GFP-HA-dally/+* |
| *Figure 2—figure supplement 2C*; *ap-Gal4, Ollas-dpp/+; UAS-dlp/+* |
| *Figure 3—figure supplement 2A*; *dlp[KO;attB]/+* |
| *Figure 3—figure supplement 2B*; *dlp[KO;attB]/dlp[KO;attB]* |
| *Figure 5—figure supplement 1A*, B, E; *dpp^{d8}/dpp^{d12}* |
| *Figure 5—figure supplement 1C*, D, F; *JAX; dpp^{d8}/dpp^{d12}* |

## Generation of transgenic flies and alleles

HA-dpp$^{\Delta N}$ plasmid was constructed by GENEWIZ by removing 21 bp encoding the 7 basic amino acids (RRPTRRK) from *HA-dpp* plasmid (***Matsuda et al., 2021***).

Ollas-HA-dpp$^{\Delta N}$ plasmid was constructed by using the *HA-dpp$^{\Delta N}$* plasmid and the *Ollas-HA-dpp* plasmid as templates. Both plasmids were digested using XhoI and BspEI as restriction enzymes. The small fragment from the *HA-dpp$^{\Delta N}$* plasmid and the large fragment from the *Ollas-HA-dpp* plasmid were used for ligation. Generation of endogenous *dpp* alleles using these plasmids were previously described (***Matsuda et al., 2021***).

## Genomic manipulation of glypicans

The *dally$^{KO}$* and *dlp$^{KO}$* alleles were generated by replacing parts of the first exon of the genes with attP cassettes. In both cases, the deleted fragment comprise part of the 5' UTR, and the complete coding sequence, including the start codon and the signal peptide encoding sequence. For the generation of *dally$^{KO}$* homology arms flanking the target region were amplified by genomic PCR using primer pairs dally_5'_for (ATA<u>GCTAGC</u>CTCTAAAGACTCAAATTAATTAATTCATTTCAGATTGCGC) and dally_5'_rev (Ata<u>catatg</u>TTTTGATGGGTGATTTCTGTGTGCAGACACAGTG), or dally_3'_for (Tat<u>actagt</u>gtaaagttcacgccatccatccgtagagttataatatcg) and dally_3'_rev (ATA<u>ggcgcgcc</u>Tttcactcaattacacgaaacagatatatattgggtacattcgc) and cloned into the 5' MCS and 3' MCS of vector pTV(Cherry) using restriction enzymes NheI and NdeI or SpeI and AscI, respectively (restriction sites underlined). The construct was subjected to P-element transgenesis and was subsequently mobilized and linearized by crossing transgenic flies to flies carrying hsFLP and hsSceI and applying heat-shock as described (***Baena-Lopez et al., 2013***).

The *dlp$^{KO}$* allele was generated by Cas9/CRISPR mediated homologous repair. Guide RNA sequences flanking the target region were gaagcaattgaagtgcaaca and aatatcggacataccgttac and were cloned into plasmid pCFD3:U6:3. The two homology arms for homologous-directed repair were amplified by genomic PCR using primer pairs dlp_5'_for (ttttCTCGA<u>GCATGC</u>tgggcatcgacacatacacatcc and dlp_5'_rev ttttGCGGCCGCatggttgtcgggtgttattaaatcgg), or dlp_3'_for (tttt<u>TCTAGA</u>tgtccgatattatataccaatggc) and dlp_3'_rev (tttt<u>CTCGAG</u>ccaacgatgctctactgtatgc) and cloned into vector pHD-DsRed-attP using restriction enzymes SphI and NotI or SpeI and XhoI, respectively. Homologous directed repair was achieved by co-injecting gRNA plasmids and the repair donor as described (***Gratz et al., 2014***).

For both *dally$^{KO}$*and *dlp$^{KO}$*, successful genomic integrations were identified by the presence of eye markers (mini-white and dsRed, respectively) followed by Cre-mediated excision of marker cassettes. The integrity of the deletion and attP integration was verified by genomic PCR and subsequent sequencing. The exact sequences at the manipulated loci are depicted in ***Figure 3—figure supplement 1***.

Dally and Dlp genomic constructs were generated in the reintegration vector pRIVwhite and incorporated into the attP site of the *dally$^{KO}$* and *dlp$^{KO}$* alleles by phiC31/attB integration (***Baena-Lopez***

*et al., 2013*). The YFP sequence in Dally constructs was inserted after lysine 90. Dally$^{\Delta HS}$ contains serine to alanine amino acid substitutions at positions 549, 569, 573, and 597. Three copies of the influenza hemagglutinin derived HA epitope were inserted after glycine 69 of Dlp.

## Immunohistochemistry

### Total staining

Wing discs from third instar larvae were dissected and stored temporarily in Phosphate Buffered Saline (PBS) (Gibco) on ice until enough samples were collected. The discs were then fixed in 4% Paraformaldehyde (PFA) in PBS for 30 min on the shaker at room temperature (25 °C). After fixation, the discs were rinsed three times quickly with PBS and three times for 15 min with PBS at 4 °C. Wing discs were permeabilized in PBST (0.3% Triton-X in PBS) and then blocked in 5% normal goat serum (NGS) in PBST for at least 30 min. Primary antibodies were added in 5% normal goat serum (NGS) in PBST for incubation over night at 4 °C. The next day, the primary antibodies were carefully removed and the samples were rinsed three times quickly in PBST and three times 15 min at room temperature in PBST. Discs were incubated in secondary antibody for 2 hr at room temperature. Afterwards the samples were washed again three times quickly and three times 15 min in PBST at room temperature. After the final washing the PBST was rinsed with PBS, then the PBS was removed completely and the samples were mounted in VECTORSHIELD on glass slides. For HS stainings, wing discs were dissected as described above. After fixation, discs were washed in PBS and blocked in 5% NGS in PBST, then treated with Heparinase III (Sigma-Aldrich) for 1.5 hr at 37 °C. Afterwards, the discs were blocked again in 5% NGS in PBST and stained as described above.

### Extracellular staining

Wing discs from third instar larvae were dissected and stored temporarily in Schneider's *Drosophila* medium (S2) on ice until enough samples were collected. The discs were then blocked in cold 5% NGS in S2 medium on ice for 10 min. The blocking solution was removed carefully and the primary antibody was added for 1 hr on ice. During the 1 hr incubation period, the tubes were tapped carefully every 15 min, to make sure the antibody is distributed evenly. After 1 hr incubation on ice, the antibody was removed and the samples were washed at least six times with cold S2 medium and another two times with cold PBS to remove excess primary antibody. Wing discs were then fixed with 4% PFA in PBS for 30 min on the shaker at room temperature (25 °C). After fixation the protocol continued as described in total stainings.

### Antibodies

Primary antibodies: rabbit-anti-phospho-Smad1/5 (41D10, Cell Signaling, #9516; 1:200), mouse-anti-Wg (4D4, DSHB, University of Iowa, RRID:AB_528512; total staining: 1:120, extracellular staining: 1:120), mouse-anti-Ptc (DSHB, University of Iowa, RRID:AB_528441; total staining: 1:40), rat-anti-HA (3F10, Roche, 11867423001, RRID:AB_390914; total staining: 1:300, extracellular staining: 1:20), rat-anti-Ollas (L2, Novus Biologicals, NBP1-06713, RRID:AB_1968650; total staining: 1:300, extracellular staining: 1:20), mouse-anti-HS (F69-3G10, amsbio; 1:100).

The following secondary antibodies were used at 1:500 dilutions in this study. Goat anti-rabbit IgG (H+L) Alexa Fluor 488 (A11008 Thermo Fischer), goat anti-rabbit IgG (H+L) Alexa Fluor 568 (A11011 Thermo Fischer), goat anti-rabbit IgG (H+L) Alexa Fluor 680 (A21109 Thermo Fischer), goat anti-rat IgG (H+L) Alexa Fluor 568 (A11077 Thermo Fischer), goat anti-mouse IgG (H+L) Alexa Fluor 568 (A11004 Thermo Fischer), goat anti-rat IgG Fc 488 (ab97089 abcam), goat anti-mouse IgG Fc Alexa Fluor 680 (115625071 Jackson Immuno Research).

### Imaging

Samples were imaged using a Leica SP5-II-MATRIX confocal microscope and Leica LAS AF and Images were analyzed using ImageJ. Figures were obtained using Omero and Adobe Illustrator.

### Quantification of pMad and extracellular staining

From each z-stack image, signal intensity profile along A/P axis was extracted from average projection of three sequential images using ImageJ (v.2.0.0-rc69/1.52 p). Each signal intensity profile collected in

Excel (Ver. 16.51) was aligned along A/P compartment boundary (based on anti-Ptc staining or pMad staining) and average signal intensity profile from different samples was generated and plotted by the script (wing_disc-alignment.py). The average intensity profile from control and experimental samples was then compared by the script (wingdisc_comparison.py). Both scripts can be obtained from GitHub (*Schmelzer, 2019*). To normalize extracellular Dpp against the level of Dally or Dally$^{\Delta HS}$, total signal intensity of extracellular Dpp staining (ExOllas staining) was divided by total GFP fluorescent signal (Dally or Dally$^{\Delta HS}$) around the Dpp producing cells in each wing disc (*Figures 4I and 5C*). The resulting signal intensity profiles (mean with SD) were generated by Prism (v.8.4.3 (471)). Figures were prepared using Omero (ver5.9.1) and Illustrator (24.1.3).

## Quantification of adult wing size

The size of each compartment was measured using ImageJ (v.2.0.0-rc69/1.52 p) and collected in Excel (Ver. 16.51). Scatter dot plots (mean with SD) were generated by Prism (v.8.4.3 (471)). Figures were prepared using Omero (ver5.9.1) and Illustrator (24.1.3).

## Statistics

All images were obtained from multiple animals (n > 3). The experiments were repeated at least two times independently with similar results. No data were excluded except if technical reason applied such as damaging samples during dissections and preparations of samples or staining failure. Statistical significance was assessed by Prism (v.8.4.3 (471)) based on the normality tests. The observed phenotypes were highly reproducible as indicated by the significance of *p* values obtained by statistical tests.

## Materials availability statement

All the materials are available from SM or GP upon request.

## Acknowledgements

We thank Markus Affolter for his continuous support during the course of this project. We thank Developmental Studies Hybridoma Bank at The University of Iowa for antibodies. Stocks obtained from the Bloomington *Drosophila* Stock Center (NIH P40OD018537) were used in this study. We thank Suzanne Eaton, Manolo Calleja, Gary Struhl, and Gustavo Aguilar for flies. We thank Etienne Schmelzer for scripts for quantification. We would like to thank Bernadette Bruno, Gina Evora, and Karin Mauro for constant and reliable supply with world's best fly food. We thank the Biozentrum Imaging Core Facility for the maintenance of microscopes and support. GP is indebted to Annette Neubüser for continuous support and to Daniela Reuter-Schmitt for excellent technical assistance. SM has been supported by the Research Fund Junior Researchers University of Basel and by an SNSF Ambizione grant (PZ00P3_180019). The work in the Pyrowolakis lab was funded by the Deutsche Forschungsgemeinschaft (DFG, German Research Foundation) under Germany's Excellence Strategy (EXC-2189-Project ID 390939984).

# Additional information

### Funding

| Funder | Grant reference number | Author |
| --- | --- | --- |
| The Swiss National Science Foundation | Ambizione grant (PZ00P3_180019) | Shinya Matsuda |
| Universität Basel | Research Fund Junior Researchers | Shinya Matsuda |
| Deutsche Forschungsgemeinschaft | Germany's Excellence Strategy (EXC-2189-Project ID 390939984) | George Pyrowolakis |

| Funder | Grant reference number | Author |
|--------|------------------------|--------|

The funders had no role in study design, data collection and interpretation, or the decision to submit the work for publication.

## Author contributions
Niklas Simon, Abu Safyan, Formal analysis, Investigation; George Pyrowolakis, Formal analysis, Supervision, Funding acquisition, Investigation, Writing – review and editing; Shinya Matsuda, Conceptualization, Formal analysis, Supervision, Funding acquisition, Investigation, Writing – original draft, Writing – review and editing

## Author ORCIDs
George Pyrowolakis (iD) http://orcid.org/0000-0002-2142-5943
Shinya Matsuda (iD) https://orcid.org/0000-0002-7541-7914

Joint Public Review: https://doi.org/10.7554/eLife.86663.3.sa1
Author Response https://doi.org/10.7554/eLife.86663.3.sa2

# Additional files

## Supplementary files
- MDAR checklist
- Supplementary file 1. Raw data and statistics to produce all the graphs in the study.

## Data availability
All data generated or analysed during this study are included in the manuscript and supporting file. Custom scripts can be obtained from GitHub (*Schmelzer, 2019*).

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
