## [Editor Report · eLife assessment]

This **important** study uses genomically-engineered glypican alleles (Dally and Dally-like) to determine the role of these proteins on the Dpp/BMP morphogen gradient in the wing disc of *Drosophila melanogaster*. The new glypican null and tagged add-back alleles, as well as a Dpp mutant that cannot bind heparan sulfate moieties in glypicans, provide **solid** results that support the model in which Dally but not Dally-like stabilizes Dpp on the cell surface by counteracting receptor-mediated Dpp internalization. This paper would be of interest to developmental biologists working on morphogens.

---

## [Referee Report · Joint Public Review]

The authors are trying to distinguish between four models of the role of glypicans (HSPGs) on the Dpp/BMP gradient in the *Drosophila* wing, schematized in Fig. 1: (1) "Restricted diffusion" (HSPGs transport Dpp via repetitive interaction of HS chains with Dpp); (2) "Hindered diffusion" (HSPGs hinder Dpp spreading via reversible interaction of HS chains with Dpp); (3) "Stabilization" (HSPGs stabilize Dpp on the cell surface via reversible interaction of HS chains with Dpp that antagonizes Tkv-mediated Dpp internalization); and (4) "Recycling" (HSPGs internalize and recycle Dpp).

To distinguish between these models, the authors generate new alleles for the glypicans Dally and Dally-like protein (Dlp) and for Dpp: a Dally knock-out allele, a Dally YFP-tagged allele, a Dally knock-out allele with 3HA-Dlp, a Dlp knock-out allele, a Dlp allele containing 3-HA tags, and a Dpp lacking the HS-interacting domain. Additionally, they use an OLLAS-tag Dpp (OLLAS being an epitope tag against which extremely high affinity antibodies exist). They examine *OLLAS-Dpp* or HA-Dpp distribution, phospho-Mad staining, adult wing size.

They find that over-expressed Dally - but not Dlp - expands Dpp distribution in the larval wing disc. They find that the Dally[KO] allele behaves like a Dally strong hypomorph Dally[MH32]. The Dally[KO] - but not the Dlp[KO] - caused reduced pMad in both anterior and posterior domains and reduced adult wing size (particularly in the Anterior-Posterior axis). These defects can be substantially corrected by supplying an endogenously tagged YFP-tagged Dally. By contrast, they were not rescued when a 3xHA Dlp was inserted in the Dally locus. These results support their conclusion that Dpp interacts with Dally but not Dlp.

They next wanted to determine the relative contributions of the Dally core or the HS chains to the Dpp distribution. To test this, they over-expressed UAS-Dally or UAS-Dally[deltaHS] (lacking the HS chains) in the dorsal wing. Dally[deltaHS] over-expression increased the distribution of *OLLAS-Dpp* but caused a reduction in pMad. They do a critical experiment, making the Dally[deltaHS] allele, they find that loss of the HS chains is nearly as severe as total loss of Dally (i.e., Dally[KO]). These results indicate that the HS are critical for Dally's role in Dpp distribution and signaling.

Prior work has shown that a stretch of 7 amino acids in the Dpp N-terminal domain is required to interact with heparin but not with Dpp receptors (Akiyama, 2008). The authors generated an HA-tagged Dpp allele lacking these residues (HA-dpp[deltaN]). It is an embryonic lethal allele, but they can get some animals to survive to larval stages if they also supply a transgene called "JAK" containing dpp regulatory sequences. In the JAK; HA-dpp[deltaN] mutant background, they find that the distribution and signaling of this Dpp molecule is largely normal. While over-expressed Dally can increase the distribution of HA-dpp[deltaN], over-expression of Dally[deltaHS] cannot. These latter results support the model that the HS chains in Dally are required for Dpp function but not because of a direct interaction with Dpp.

In the last part of the results, they attempt to determine if the Dpp receptor Thickveins (Tkv) is required for Dally-HS chains interaction. The 2008 (Akiyama) model posits that Tkv activates pMad downstream of Dpp and also internalizes and degrades Dpp. A 2022 (Romanova-Michaelides) model proposes that Dally (not Tkv) internalizes Dpp. To distinguish between these models, the authors deplete Tkv from the dorsal compartment of the wing disc and found that extracellular Dpp increased and expanded in that domain. These results support the model that Tkv is required to internalize Dpp. They then tested the model that Dally antagonizes Tkv-mediated Dpp internalization by determining whether the defective extracellular Dpp distribution in Dally[KO] mutants could be rescued by depleting Tkv. Extracellular Dpp did increase in the D vs V compartment, potentially providing some support for their model. The results are statistically significant but the statistics are buried in an excel file without a read-me page. The code for the statistics is available from Github. These p values should be made more readily accessible and/or intelligible to the reader.

Strengthens:

1. New genomically-engineered alleles

A considerable strength of the study is the generation and characterization of new Dally, Dlp and Dpp alleles. These reagents will be of great use to the field.

2. Surveying multiple phenotypes

The authors survey numerous parameters (Dpp distribution, Dpp signaling (pMad) and adult wing phenotypes) which provides many points of analysis.

Weaknesses (minor):

1. The results are statistically significant but the statistics are buried in a dense excel file without a read-me page. The code for the statistics is available from Github. These p values should be made more readily accessible to the reader.

An appraisal of whether the authors achieved their aims, and whether the results support their conclusions.

The authors' model is that Dally (not Dlp) is required for Dpp distribution and signaling but that this is not due to a direct interaction with Dpp. Rather, they posit that Dally-HS antagonize Tkv-mediated Dpp internalization. Currently the results of the experiments could be considered consistent with their model. Finally, their results support the idea that one or more as-yet unidentified proteins interact with Dally-HS chains to control Dpp distribution and signaling in the wing disc.

There is much debate and controversy in the Dpp morphogen field. The generation of new, high quality alleles in this study will be useful to *Drosophila* community, and the results of this study support the concept that Tkv but not Dally regulate Dpp internalization. Thus the work could be impactful and fuel new debates among the morphogen researchers.

---

## [Author Response]

The following is the authors’ response to the current reviews.

We agree with the reviewer that the statistics are buried in a dense excel file without a read-me page. We will address this by making a summary excel page for p-values during the production process.

The following is the authors’ response to the original reviews.

**eLife assessment**
This important study uses genomically-engineered glypican alleles to demonstrate convincingly that Dally (not Dally-like protein [Dlp]) is the key contributor to formation of the Dpp/BMP morphogen gradient in the wing disc of *Drosophila*. The authors provide solid genetic evidence that, surprisingly, the core domain of Dally appears to suffice to trap Dpp at the cell surface. They conclude with a model according to which Dally modulates the range of Dpp signaling by interfering with Dpp's internalization by the Dpp receptor Thickveins.
**Public Reviews:**

**Reviewer #1 (Public Review):**
How morphogens spread within tissues remains an important question in developmental biology. Here the authors revisit the role of glypicans in the formation of the Dpp gradient in wing imaginal discs of *Drosophila*. They first use sophisticated genome engineering to demonstrate that the two glypicans of *Drosophila* are not equivalent despite being redundant for viability. They show that Dally is the relevant glypican for Dpp gradient formation. They then provide genetic evidence that, surprisingly, the core domain of Dally suffices to trap Dpp at the cell surface (suggesting a minor role for GAGs). They conclude with a model that Dally modulates the range of Dpp signaling by interfering with Dpp's degradation by Tkv. These are important conclusions, but more independent (biochemical/cell biological) evidence is needed.As indicated above, the genetic evidence for the predominant role of Dally in Dpp protein/signalling gradient formation is strong. In passing, the authors could discuss why overexpressed Dlp has a negative effect on signaling, especially in the anterior compartment. The authors then move on to determine the role of GAG (=HS) chains of Dally. They find that in an overexpression assay, Dally lacking GAGs traps Dpp at the cell surface and, counterintuitively, suppresses signaling (fig 4 C, F). Both findings are unexpected and therefore require further validation and clarification, as outlined in a and b below.a. In loss of function experiments (dallyDeltaHS replacing endogenous dally), Dpp protein is markedly reduced (fig 4R), as much as in the KO (panel Q), suggesting that GAG chains do contribute to trapping Dpp at the cell surface. This is all the more significant that, according to the overexpression essays, DallyDeltaHS seems more stable than WT Dally (by the way, this difference should also be assessed in the knock-ins, which is possible since they are YFP-tagged). The authors acknowledge that HS chains of Dally are critical for Dpp distribution (and signaling) under physiological conditions. If this is true, one can wonder why overexpressed dally core 'binds' Dpp and whether this is a physiologically relevant activity.

According to the overexpression assay, DallyDeltaHS seems more stable than WT Dally (Fig. 4B’, E’, 5A’, B’). As the reviewer suggested, we addressed the difference using the two knock-in alleles and found that DallyDeltaHS is more stable than WT Dally (Fig.4 L, M inset), further emphasizing the insufficient role of core protein of Dally for extracellular Dpp distribution.

In summary, we showed that, although Dally interacts with Dpp mainly through its core protein from the overexpression assay (Fig. 4E, I), HS chains are essential for extracellular Dpp distribution (Fig. 4R). Thus, the core protein of Dally alone is not sufficient for extracellular Dpp distribution under physiological conditions. These results raise a question about whether the interaction of core protein of Dally with Dpp is physiologically relevant. Since the increase of HS upon dally expression but not upon dlp expression resulted in the accumulation of extracellular Dpp (Figure 2) and this accumulation was mainly through the core protein of Dally (Fig. 4E, I), we speculate that the interaction of the core protein of Dally with Dpp gives ligand specificity to Dally under physiological conditions.

To understand the importance of the interaction of core protein of Dally with Dpp under physiological conditions, it is important to identify a region responsible for the interaction. Our preliminary results overexpressing a dally mutant lacking the majority of core protein (but keeping the HS modified region intact) showed that HS chains modification was also lost. Although this is consistent with our results that enzymes adding HS chains also interact with the core protein of Dally (Fig. 4D), the dally mutant allele lacking the core protein would hamper us from distinguishing the role of core protein of Dally from HS chains.

Nevertheless, we can infer the importance of the interaction of core protein of Dally with Dpp using dally[3xHA-dlp, attP] allele, where dlp is expressed in dally expressing cells. Since Dally-like is modified by HS chains but does not interact with Dpp (Fig. 2, 4), dally[3xHA-dlp, attP] allele mimics a dally allele where HS chains are properly added but interaction of core protein with Dpp is lost. As we showed in Fig.3O, S, the allele could not rescue dallyKO phenotypes, consistent with the idea that interaction of core protein of Dally with Dpp is essential for Dpp distribution and signaling and HS chain alone is not sufficient for Dpp distribution.

b. Although the authors' inference that dallycore (at least if overexpressed) can bind Dpp. This assertion needs independent validation by a biochemical assay, ideally with surface plasmon resonance or similar so that an affinity can be estimated. I understand that this will require a method that is outside the authors' core expertise but there is no reason why they could not approach a collaborator for such a common technique. In vitro binding data is, in my view, essential.

We agree with the reviewer that a biochemical assay such as SPR helps us characterize the interaction of core protein of Dally and Dpp (if the interaction is direct), although the biochemical assay also would not demonstrate the interaction under the physiological conditions.

However, SPR has never been applied in the case of Dpp, probably because purifying functional refolded Dpp dimer from bacteria has previously been found to be stable only in low pH and be precipitated in normal pH buffer (Groppe J, et al., 1998)(Matsuda et al., 2021). As the reviewer suggests, collaborating with experts is an important step in the future.

Nevertheless, SPR was applied for the interaction between BMP4 and Dally (Kirkpatrick et al., 2006), probably because BMP4 is more stable in the normal buffer. Although the binding affinity was not calculated, SPR showed that BMP4 directly binds to Dally and this interaction was only partially inhibited by molar excess of exogenous HS, suggesting that BMP4 can interact with core protein of Dally as well as its HS chains. In addition, the same study applied Co-IP experiments using lysis of S2 cells and showed that Dpp and core protein of Dally are co-immunoprecipitated, although it does not demonstrate if the interaction is direct.

In a subsequent set of experiments, the authors assess the activity of a form of Dpp that is expected not to bind GAGs (DppDeltaN). Overexpression assays show that this protein is trapped by DallyWT but not dallyDeltaHS. This is a good first step validation of the deltaN mutation, although, as before, an invitro binding assay would be preferable.

Our overexpression assays actually showed that DppDeltaN is trapped by DallyWT and by dallyDeltaHS at similar levels (Fig. 5C), indicating that interaction of DppDeltaN and HS chains of Dally is largely lost but DppDeltaN can still interact with core protein of Dally.

We thank the reviewer for the suggesting the in vitro experiment. Although we decided not to develop biophysical experiments such as SPR for Dpp in this study due to the reasons discussed above, we would like to point out that our result is consistent with a previous Co-IP experiment using S2 cells showing that DppDeltaN loses interaction with heparin (Akiyama2008).

However, in contrast to our results, the same study also proposed by Co-IP experiments using S2 cells that DppDeltaN loses interaction with Dally (Akiyama2008). Although it is hard to conclude since western blotting was too saturated without loading controls and normalization (Fig. 1C in Akiyama 2008), and negative in vitro experiments do not necessarily demonstrate the lack of interaction in vivo. One explanation why the interaction was missed in the previous study is that some factors required for the interaction of DppDeltaN with core protein of Dally are missing in S2 cells. In this case, in vivo interaction assay we used in this study has an advantage to robustly detect the interaction.

Nevertheless, the authors show that DppDeltaN is surprisingly active in a knock-in strain. At face value (assuming that DeltaN fully abrogates binding to GAGs), this suggests that interaction of Dpp with the GAG chains of Dally is not required for signaling activity. This leads to authors to suggest (as shown in their final model) that GAG chains could be involved in mediating the interactions of Dally with Tkv and not with Dpp. This is an interesting idea, which would need to be reconciled with the observation that the distribution of Dpp is affected in dallyDeltaHS knock-ins (item a above). It would also be strengthened by biochemical data (although more technically challenging than the experiments suggested above). In an attempt to determine the role of Dally (GAGs in particular) in the signaling gradient, the paper next addresses its relation to Tkv. They first show that reducing Tkv leads to Dpp accumulation at the cell surface, a clear indication that Tkv normally contributes to the degradation of Dpp. From this they suggest that Tkv could be required for Dpp internalisation although this is not shown directly. The authors then show that a Dpp gradient still forms upon double knockdown (Dally and Tkv). This intriguing observation shows that Dally is not strictly required for the spread of Dpp, an important conclusion that is compatible with early work by Lander suggesting that Dpp spreads by free diffusion. These result show that Dally is required for gradient formation only when Tkv is present. They suggest therefore that Dally prevents Tkv-mediated internalisation of Dpp. Although this is a reasonable inference, internalisation assays (e.g. with anti-Ollas or anti-HA Ab) would strengthen the authors' conclusions especially because they contradict a recent paper from the Gonzalez-Gaitan lab.

Thanks for suggesting the internalization assay. As we discussed in the discussion, our results suggest that extracellular Dpp distribution is severely reduced in dally mutants due to Tkv mediated internalization of Dpp (Fig. 6). Thus, extracellular Dpp available for labelling with nanobody is severely reduced in dally mutants, which can explain the reduced internalization of Dpp in dally mutants in the internalization assay. Therefore, we think that the nanobody internalization assay would not distinguish the two contradicting possibilities.

The paper ends with a model suggesting that HS chains have a dual function of suppressing Tkv internalisation and stimulating signaling. This constitutes a novel view of a glypican's mode of action and possibly an important contribution of this paper. As indicated above, further experiments could considerably strengthen the conclusion. Speculation on how the authors imagine that GAG chains have these activities would also be warranted.

Thank you very much!

**Reviewer #2 (Public Review):**
The authors are trying to distinguish between four models of the role of glypicans (HSPGs) on the Dpp/BMP gradient in the *Drosophila* wing, schematized in Fig. 1: (1) "Restricted diffusion" (HSPGs transport Dpp via repetitive interaction of HS chains with Dpp); (2) "Hindered diffusion" (HSPGs hinder Dpp spreading via reversible interaction of HS chains with Dpp); (3) "Stabilization" (HSPGs stabilize Dpp on the cell surface via reversible interaction of HS chains with Dpp that antagonizes Tkv-mediated Dpp internalization); and (4) "Recycling" (HSPGs internalize and recycle Dpp).To distinguish between these models, the authors generate new alleles for the glypicans Dally and Dally-like protein (Dlp) and for Dpp: a Dally knock-out allele, a Dally YFP-tagged allele, a Dally knock-out allele with 3HA-Dlp, a Dlp knock-out allele, a Dlp allele containing 3-HA tags, and a Dpp lacking the HS-interacting domain. Additionally, they use an OLLAS-tag Dpp (OLLAS being an epitope tag against which extremely high affinity antibodies exist). They examine *OLLAS-Dpp* or HA-Dpp distribution, phospho-Mad staining, adult wing size.They find that over-expressed Dally - but not Dlp - expands Dpp distribution in the larval wing disc. They find that the Dally[KO] allele behaves like a Dally strong hypomorph Dally[MH32]. The Dally[KO] - but not the Dlp[KO] - caused reduced pMad in both anterior and posterior domains and reduced adult wing size (particularly in the Anterior-Posterior axis). These defects can be substantially corrected by supplying an endogenously tagged YFP-tagged Dally. By contrast, they were not rescued when a 3xHA Dlp was inserted in the Dally locus. These results support their conclusion that Dpp interacts with Dally but not Dlp.They next wanted to determine the relative contributions of the Dally core or the HS chains to the Dpp distribution. To test this, they over-expressed UAS-Dally or UAS-Dally[deltaHS] (lacking the HS chains) in the dorsal wing. Dally[deltaHS] over-expression increased the distribution of *OLLAS-Dpp* but caused a reduction in pMad. Then they write that after they normalize for expression levels, they find that Dally[deltaHS] only mildly reduces pMad and this result indicates a major contribution of the Dally core protein to Dpp stability.

Thanks for the comments. We actually showed that compared with Dally overexpression, Dally[deltaHS] overexpression only mildly reduces extracellular Dpp accumulation (Fig. 4I). This indicates a major contribution of the Dally core protein to interaction with Dpp, although the interaction is not sufficient to sustain extracellular Dpp distribution and signaling gradient.

The "normalization" is a key part of this model and is not mentioned how the normalization was done. When they do the critical experiment, making the Dally[deltaHS] allele, they find that loss of the HS chains is nearly as severe as total loss of Dally (i.e., Dally[KO]). Additionally, experimental approaches are needed here to prove the role of the Dally core.

Since the expression level of Dally[deltaHS] is higher than Dally when overexpressed, we normalized extracellular Dpp distribution (a-Ollas staining) against GFP fluorescent signal (Dally or Dally[deltaHS]). To do this, we first extracted both signal along the A-P axis from the same ROI in the previous version. The ratio was calculated by dividing the intensity of a-Ollas staining with the intensity of GFP fluorescent signal at a given position x. The average profile from each normalized profile was generated and plotted using the script described in the method (wingdisc_comparison.py) as other pMad or extracellular staining profiles.

Although this analysis provides normalized extracellular Dpp accumulation at different positions along the A-P axis, we are more interested in the total amount of Dpp or DppDeltaN accumulation upon Dally or dallyDeltaHS expression. Therefore, in the revised ms, we decided to normalize total amount of extracellular Dpp against the level of Dally or Dally[deltaHS] by dividing total signal intensity of extracellular Dpp staining (ExOllas staining) by total GFP fluorescent signal (Dally or Dally[deltaHS]) around the Dpp producing cells in each wing disc. Statistical analysis showed that accumulation of extracellular Dpp is only slightly reduced without HS chains (Fig.4I), indicating that Dally interacts with Dpp mainly through its core protein.

We agree with the reviewer that additional experimental approaches are needed to address the role of the core protein of Dally. As we discussed in the response to the reviewer1, to understand the importance of the interaction of core protein of Dally with Dpp, it is important to identify a region responsible for the interaction. Our preliminary results overexpressing a dally mutant lacking the majority of core protein (but keeping the HS modified region intact) showed that HS chains modification was also lost. Although this is consistent with our results that enzymes adding HS chains also interact with the core protein of Dally (Fig. 4D), the dally mutant allele lacking the core protein would hamper us from distinguishing the role of the core protein of Dally from HS chains.

Nevertheless, we can infer the importance of the interaction of core protein of Dally with Dpp using dally[3xHA-dlp, attP] allele, where dlp is expressed in dally expressing cells. Since Dally-like is modified by HS chains but does not interact with Dpp (Fig. 2, 4), dally[3xHA-dlp, attP] allele mimics a dally allele where HS chains are properly added but interaction of core protein with Dpp is lost. As we showed in Fig.3O, S, the allele could not rescue dallyKO phenotypes, consistent with the idea that interaction of core protein of Dally with Dpp is essential for Dpp distribution and signaling.

Prior work has shown that a stretch of 7 amino acids in the Dpp N-terminal domain is required to interact with heparin but not with Dpp receptors (Akiyama, 2008). The authors generated an HA-tagged Dpp allele lacking these residues (HA-dpp[deltaN]). It is an embryonic lethal allele, but they can get some animals to survive to larval stages if they also supply a transgene called “JAX” containing dpp regulatory sequences. In the JAX; HA-dpp[deltaN] mutant background, they find that the distribution and signaling of this Dpp molecule is largely normal. While over-expressed Dally can increase the distribution of HA-dpp[deltaN], over-expression of Dally[deltaHS] cannot. These latter results support the model that the HS chains in Dally are required for Dpp function but not because of a direct interaction with Dpp.

Our overexpression assays actually showed that both Dally and Dally[deltaHS] can accumulate Dpp upon overexpression and the accumulation of Dpp is comparable after normalization (Fig. 5C), consistent with the idea that interaction of DppdeltaN and HS chains are largely lost. As the reviewer pointed out, these results support the model that the HS chains in Dally are required for Dpp function but not because of a direct interaction with Dpp.

In the last part of the results, they attempt to determine if the Dpp receptor Thickveins (Tkv) is required for Dally-HS chains interaction. The 2008 (Akiyama) model posits that Tkv activates pMad downstream of Dpp and also internalizes and degrades Dpp. A 2022 (Romanova-Michaelides) model proposes that Dally (not Tkv) internalizes Dpp.To distinguish between these models, the authors deplete Tkv from the dorsal compartment of the wing disc and found that extracellular Dpp increased and expanded in that domain. These results support the model that Tkv is required to internalize Dpp.They then tested the model that Dally antagonizes Tkv-mediated Dpp internalization by determining whether the defective extracellular Dpp distribution in Dally[KO] mutants could be rescued by depleting Tkv. Extracellular Dpp did increase in the D vs V compartment, potentially providing some support for their model. However, there are no statistics performed, which is needed for full confidence in the results. The lack of statistics is particularly problematic (1) when they state that extracellular Dpp does not rise in ap>tkv RNAi vs ap>tkv RNAi, dally[KO] wing discs (Fig. 6E) or (2) when they state that extracellular Dpp gradient expanded in the dorsal compartment when tkv was dorsally depleted in dally[deltaHS] mutants (Fig. 6I). These last two experiments are important for their model but the differences are assessed only visually. In fact, extracellular Dpp in ap>tkv RNAi, dally[KO] (Fig. 6B) appears to be lower than extracellular Dpp in ap>tkv RNAi (Fig. 6A) and the histogram of Dpp in ap>tkv RNAi, dally[KO] is actually a bit lower than Dpp in ap>tkv RNAi, But the author claim that there is no difference between the two. Their conclusion would be strengthened by statistical analyses of the two lines.

We provided statistics for all the quantifications for pMad and extracellular Dpp distribution as supplementary data. In the previous version, we argued that extracellular Dpp level in ap>tkvRNAi, dallyKO (Fig.6B) does not increase compared with that in ap>tkvRNAi (Fig.6A). Statistical analysis (t-test) showed that the extracellular Dpp level in Fig. 6B is similar to or lower than that in Fig. 6A (Fig. 6E), confirming our conclusion. Statistical analysis (t-test) also confirmed that extracellular Dpp distribution expanded when tkv was knocked down in dallyHS mutants (Fig. 6I).

Strengths:1. New genomically-engineered allelesA considerable strength of the study is the generation and characterization of new Dally, Dlp and Dpp alleles. These reagents will be of great use to the field.

Thanks. We hope that these resources are indeed useful to the field.

2. Surveying multiple phenotypesThe authors survey numerous parameters (Dpp distribution, Dpp signaling (pMad) and adult wing phenotypes) which provides many points of analysis.

Thanks!

Weaknesses:1. Confusing discussion regarding the Dally core vs HS in Dpp stability. They don't provide any measurements or information on how they "normalize" for the level of Dally vs Dally[deltaHS]? This is important part of their model that currently is not supported by any measurements.

We explained how we normalized in the above section and updated the method section in the revised ms.

2. Lacking quantifications and statistical analyses:a. Why are statistical significance for histograms (pMad and Dpp distribution) not supplied? These histograms provide the key results supporting the authors' conclusions but no statistical tests/results are presented. This is a pervasive shortcoming in the current study.

Thanks. We provided t-test analyses together with the raw data as supplementary data.

b. dpp[deltaN] with JAX transgene - it would strengthen the study to supply quantitative data on the percent survival/lethal stage of dpp[deltaN] mutants with or without the JAK transgene

In this study, we are interested in the role of dpp[deltaN] during the wing disc development. Therefore, we decided not to perform the detailed analysis on the percent survival/lethal stage of dpp[deltaN] mutants with or without the JAX transgene in the current study. Nevertheless, the fact that dpp[deltaN] allele is maintained with a balanced stock and JAX;dpp[deltaN] allele can be maintained as homozygous stock indicates that the lethality of dpp[deltaN] allele comes from the early stages. Indeed, our preliminary results showed that pMad signal is severely lost in the dpp[deltaN] embryo without JAX (data not shown), indicating that the allele is lethal at early embryonic stages.

c. The graphs on wing size etc should start at zero.

Thanks. We corrected this in the current ms.

d. The sizes of histograms and graphs in each figure should be increased so that the reader can properly assess them. Currently, they are very small.

Thanks. We changed the sizes in the current ms.

The authors' model is that Dally (not Dlp) is required for Dpp distribution and signaling but that this is not due to a direct interaction with Dpp. Rather, they posit that Dally-HS antagonize Tkv-mediated Dpp internalization. Currently the results of the experiments could be considered consistent with their model, but as noted above, the lack of statistical analyses of some parameters is a weakness.

Thanks. We now performed and provided the statistical analyses in the revised ms.

One problematic part of their result for me is the role of the Dally core protein (Fig. 7B). There is a mis-match between the over-expression results and Dally allele lacking HS (but containing the core). Finally, their results support the idea that one or more as-yet unidentified proteins interact with Dally-HS chains to control Dpp distribution and signaling in the wing disc.

Our results simply suggest that Dpp can interact with Dally mainly through core protein but this interaction is not sufficient to sustain extracellular Dpp gradient formation under physiological conditions (dallyDeltaHS) (Fig. 4Q). We find that the mis-match is not problematic if the role of Dally is not simply mediated through interaction with Dpp. We speculate that interaction of Dpp and core protein of Dally is transient and not sufficient to sustain the Dpp gradient without HS chains of Dally stabilizing extracellular Dpp distribution by blocking Tkv-mediated Dpp internalization.

There is much debate and controversy in the Dpp morphogen field. The generation of new, high quality alleles in this study will be useful to *Drosophila* community, and the results of this study support the concept that Tkv but not Dally regulate Dpp internalization. Thus the work could be impactful and fuel new debates among morphogen researchers.

Thanks.

The manuscript is currently written in a manner that really is only accessible to researchers who work on the Dpp gradient. It would be very helpful for the authors to re-write the manuscript and carefully explain in each section of the results (1) the exact question that will be asked, (2) the prior work on the topic, (3) the precise experiment that will be done, and (4) the predicted results. This would make the study more accessible to developmental biologists outside of the morphogen gradient and *Drosophila* communities.

Thanks. We modified texts and changed the order of Fig.5. We hope that the changes make this study more accessible to developmental biologists outside of the field.